# SKTFORMER: A SKELETON TRANSFORMER FOR LONG SEQUENCE DATA

## ABSTRACT

Transformers have become a preferred tool for modeling sequential data. Many studies of using Transformers for long sequence modeling focus on reducing computational complexity. They usually exploit the low-rank structure of data and approximate a long sequence by a sub-sequence. One challenge with such approaches is how to make an appropriate balance between information preservation and noise reduction: the longer the sub-sequence used to approximate the long sequence, the better the information is preserved but at a price of introducing more noise into the model and of course more computational costs. We propose skeleton transformer, SKTformer for short, an efficient transformer architecture that improves upon the previous attempts to negotiate this tradeoff. It introduces two mechanisms to effectively reduce the impact of noise while still keeping the computation linear to the sequence length: a smoothing block to mix information over long sequences and a matrix sketch method that simultaneously selects columns and rows from the input matrix. We verify the effectiveness of SKTformer both theoretically and empirically. Extensive studies over both Long Range Arena (LRA) datasets, and six time-series forecasting show that SKTformer significantly outperforms both vanilla Transformer and other state-of-the-art variants of Transformer. Code is available at https://anonymous.4open.science/r/SKTFormer-B33B/

## 1 INTRODUCTION

Transformer type models (Vaswani et al., 2017) have achieved many breakthroughs in various artificial intelligence areas, such as natural language processing (NLP) (Brown et al., 2020; Clark et al., 2020; Devlin et al., 2018; Liu et al., 2019), computer vision (CV) (Dosovitskiy et al., 2020; Liu et al., 2021; Touvron et al., 2021; Yuan et al., 2021; Zhou et al., 2021b), and time series forecasting (Xu et al., 2021; Zhou et al., 2022). The self-attention scheme plays a key role in those transformer-based models, which efficiently capture long-term global and short-term local correlations when the length of the token sequence is relatively small. Due to the quadratic complexity of standard self-attention, many approaches have been developed to reduce the computational complexity of Transformer for long sequences (e.g., (Zhu et al., 2021)). Most of them try to exploit the special patterns of attention matrix, such as low-rankness, locality, sparsity, or graph structures. One group of approaches is to build a linear approximation for the $\mathrm{softmax}$ operator (e.g., (Chen et al., 2021; Choromanski et al., 2020; Chowdhury et al., 2021; Qin et al., 2021)). Despite the efficiency of the linear approximation, these approximation methods often perform worse than the original $\mathrm{softmax}$ based attention. More discussion of efficient transformer for long sequence can be found in the section of related work.

In this work, we will focus on approaches that assume a low-rank structure of input matrix. They approximate the global information in a long sequence by a sub-sequence (i.e., short sequence) of landmarks, and only compute attention between queries and selected landmarks (e.g., (Ma et al., 2021; Nguyen et al., 2021; Zhu et al., 2021; Zhu & Soricut, 2021)). Although those models enjoy linear computational cost and often better performance than vanilla Transformer, they face one major challenge, i.e., how to balance between information preserving and noise reduction. By choosing a larger number of landmarks, we are able to preserve more global information but at the price of introducing more noise into the sequential model and more computational cost.

In this work, we propose an efficient Transformer architecture, termed Skeleton Transformer, or SKTformer for short, that introduces two mechanisms to explicitly address the balance. First,

we introduce a smoothing block into the Transformer architecture. It effectively mixes global information over the long sequence by the Fourier analysis and local information over the sequence by a convolution kernel. Through the information mixture, we are able to reduce the noise for individual tokens over the sequence, and at the same time, improve their representativeness for the entire sequence. Second, we introduce a matrix sketch technique to approximate the input matrix by a smaller number of rows and columns. A standard self-attention can be seen as reweighing the columns of the value matrix. Important columns are assigned high attention weights and remain in the output matrix, while small attention weights eliminate insignificant columns. The self-attention mechanism is equivalent to column selection if we replace the $\mathrm{softmax}$ operator with the corresponding $\mathrm{argmax}$ operator. However, sampling only columns may not generate a good summary of the matrix, and could be subjected to noises to individual columns. We address this problem by exploiting CUR (Drineas et al., 2008) or Skeleton approximation technique (Chiu & Demanet, 2013) in the matrix approximation community. Theoretically, for a rank-$r$ matrix $\boldsymbol{X} \in \mathbb{R}^{n \times d}$, we can take $\mathcal{O}(r \log d)$ column samples and $\mathcal{O}(r \log n)$ row samples to construct a so-called Skeleton approximation $\boldsymbol{X} \approx \boldsymbol{C}\boldsymbol{U}\boldsymbol{R}$, where $\boldsymbol{C}$ and $\boldsymbol{R}$ are matrices consisting of the columns and rows of $\boldsymbol{X}$, respectively, and $\boldsymbol{U}$ is the pseudo-inverse of their intersection. By combing these mechanism, we found, both theoretically and empirically, that SKTformer is able to preserve global information over long sequence and reduce the impact of noise simultaneously, thus leading to better performance than state-of-the-art variants of Transformer for long sequences, without having to sacrifice the linear complexity w.r.t. sequence length.

In short, we summarize our main contributions as follows:

1. We propose a Skeleton Transformer (SKTformer), an efficient model that integrates a smoother, column attention and row attention components to unfold a randomized linear matrix sketch algorithm.

2. By randomly selecting a fixed number of rows and columns, the proposed model achieves near-linear computational complexity and memory cost. The effectiveness of this selection method is verified both theoretically and empirically.

3. We conduct extensive experiments over Long-term sequence, long-term time series forecasting and GLUE tasks. In particular, the Long Range Arena benchmark (Tay et al., 2021), achieves an average accuracy of 64% and 66% with fixed parameters (suggested setting in Mathieu et al. (2014); Tay et al. (2021)) and fine-tuned parameters respectively. It improves from 62% of the best transformer-type model. Moreover, it also has a comparable performance with the recent state-of-art long-term time series forecasting models for long-term time series forecasting and GLUE tasks

**Organization.** We structure the rest of this paper as follows: In Section 2, we briefly review the relevant literature on efficient transformers and Skeleton approximations. Section 3 introduces the model structure and performs a theoretical analysis to justify the proposed model. We empirically verify the efficiency and accuracy of SKTformer in Section 4. we discuss limitations and future directions in Section 5. Technical proofs and experimental details are provided in the appendix.

## 2 RELATED WORK

This section provides an overview of the literature focusing on efficient Transformer models. The techniques include sparse or local attention, low-rankness, and kernel approximation. We refer the reader interested in their details to the survey (Tay et al., 2020c).

**Sparse Attention.** The general idea of these methods is restricting the query token to perform attention only within a specific small region, such as its local region or some global tokens. In this setting, the attention matrix becomes sparse compared to the original one. (Qiu et al., 2019) proposes BlockBert, which introduces sparse block structures into the attention matrix by multiplying a masking matrix. (Parmar et al., 2018) applies self-attention within blocks for the image generation task. (Liu et al., 2018) divides a sequence into blocks and uses a stride convolution to reduce the model complexity. However, these block-type Transformers ignore the connections among blocks. To address this issue, Transformer-XL (Dai et al., 2019) and Compressive Transformer (Rae et al., 2019) propose a recurrence mechanism to connect multiple blocks. Transformer-LS (Zhu et al., 2021) combines local attention with a dynamic projection to capture long-term dependence. (Tay et al.,

2020b) uses a meta-sorting network to permute over sequences and quasi-global attention with local windows to improve memory efficiency.

Another approach in this category is based on stride attention. Longformer (Beltagy et al., 2020) uses dilated sliding windows to obtain a sparse attention matrix. Sparse Transformers (Child et al., 2019) consider approximating a dense attention matrix by several sparse factorization methods. In addition, some methods reduce the complexity by clustering tokens. For example, Reformer (Kitaev et al., 2020b) uses a hash similarity measure to cluster tokens, and Routing Transformer (Roy et al., 2021) uses k-means to cluster tokens. BigBird (Zaheer et al., 2020) proposes a generalized attention mechanism described by a directed graph to reduce attention complexity. (Lee-Thorp et al., 2021) considers using 2D Fourier Transformation to mix the token matrix directly. (Tan et al., 2021) uses max pooling scheme to reduce the computation costs.

**Low-rank and Kernel Methods.** Inducing low-rankness into the attention matrix can quickly reduce the complexity and the kernel approximation is widely applied in efficient low-rank approximation. Linformer (Wang et al., 2020) and Luna (Ma et al., 2021) approximate $\mathrm{softmax}$ with linear functions, which yield a linear time and space complexity. (Choromanski et al., 2020) and (Peng et al., 2021) use random features tricks and reach promising numerical performance. (Winata et al., 2020) proposes Low-Rank Transformer based on matrix factorization. FMMformer (Nguyen et al., 2021) combines the fast multipole method with the kernel method. Synthesizer (Tay et al., 2020a) uses a random low-rank matrix to replace the attention matrix. Nyströmformer (Xiong et al., 2021) adopts the Nyström method to approximate standard self-attention. Linear Transformer (Katharopoulos et al., 2020) expresses self-attention as a linear dot-product of kernel feature maps. (Zhu & Soricut, 2021) applies the Multigrid method to efficiently compute the attention matrix recursively. Cosformer (Qin et al., 2021) develops a cosine-based re-weighting mechanism to linearize the $\mathrm{softmax}$ function. (Chen et al., 2021) proposes the Scatterbrain, which unifies locality-sensitive hashing and the kernel method into attention for accurate and efficient approximation.

## 3 SKTFORMER

We start by going over the vanilla attention. For a sequence of length $n$, the vanilla self-attention in the transformer is dot-product type (Vaswani et al., 2017). Following standard notation, the attention matrix $\boldsymbol{A} \in \mathbb{R}^{n \times n}$ is defined as:

$$\boldsymbol{A} = \mathrm{softmax}\left(\frac{1}{\sqrt{d}}\boldsymbol{Q}\boldsymbol{K}^{\top}\right), \tag{1}$$

where $\boldsymbol{Q} \in \mathbb{R}^{n \times d}$ denotes the queries while $\boldsymbol{K} \in \mathbb{R}^{n \times d}$ denotes the keys, and $d$ represents the hidden dimension. By multiplying the attention weights $\boldsymbol{A}$ with the values $\boldsymbol{V} \in \mathbb{R}^{n \times d}$, we can calculate the new values as $\hat{\boldsymbol{V}} = \boldsymbol{A}\boldsymbol{V}$.

Intuitively, the attention is the weighted average over the old ones, where the weights are defined by the attention matrix $\boldsymbol{A}$. In this paper, we consider generating $\boldsymbol{Q}$, $\boldsymbol{K}$ and $\boldsymbol{V}$ via the linear projection of the input token matrix $\boldsymbol{X}$:

$$\boldsymbol{Q} = \boldsymbol{X}\boldsymbol{W}_Q, \ \boldsymbol{K} = \boldsymbol{X}\boldsymbol{W}_K, \ \boldsymbol{V} = \boldsymbol{X}\boldsymbol{W}_V,$$

where $\boldsymbol{X} \in \mathbb{R}^{n \times d}$ and $\boldsymbol{W}_Q, \boldsymbol{W}_K, \boldsymbol{W}_V \in \mathbb{R}^{d \times d}$.

The vanilla procedure has two drawbacks in concentrating the information from $\boldsymbol{V}$. First, when computing the $\boldsymbol{Q}\boldsymbol{K}^{\top}$ part, full dense matrix multiplication is involved at a cost of $\mathcal{O}(n^2)$ vector multiplications. It can be prohibitive for long sequence problems. On the other hand, if we view the $\mathrm{softmax}$ operator as an approximation of the argmax counterpart, $\hat{\boldsymbol{V}}$ becomes a row selection from $\boldsymbol{V}$. This column-wise information concentration is ignored.

### 3.1 SKELETON ATTENTION

We propose a Skeleton self-attention structure motivated by the Skeleton approximation to address those issues. First, we modify the original self-attention to build the column self-attention as follows:

$$\hat{\boldsymbol{V}}_1 = \mathrm{softmax}\left(\frac{1}{\sqrt{d}}\boldsymbol{Q}\boldsymbol{K}^{\top}\boldsymbol{P}_1^{\top}\right)\boldsymbol{P}_1\boldsymbol{V},$$

where $\boldsymbol{P}_1 \in \mathbb{R}^{s_1 \times n}$ denotes the sampling matrix and $s_1$ is the number of columns sampled. Let $i_1 < i_2 < ... < i_{s_1}$ be the indices of the randomly sampled columns. Let $\boldsymbol{P}_{1,ab}$ denote the element located at the $a$-th column and $b$-th row and we have $\boldsymbol{P}_{1,ab} = 1$ if $i_a = b$ and $0$ otherwise. By these constructions, we can reduce the computational cost to $\mathcal{O}(ns_1d^2 + ns_1^2d)$.

Similarly, we build the row sampling matrix $\boldsymbol{P}_2 \in \mathbb{R}^{d \times s_2}$ indicating the locations of the $s_2$ sample rows. Compute the row self-attention as:

$$\hat{\boldsymbol{V}}_2 = \boldsymbol{V} \boldsymbol{P}_2 \, \mathrm{softmax}\left( \frac{1}{\sqrt{n}} \boldsymbol{P}_2^\top \boldsymbol{K}^\top \boldsymbol{Q} \right).$$

Finally, we apply the layer-norm on $\hat{\boldsymbol{V}}_1$ and $\hat{\boldsymbol{V}}_2$ and then them together to generate the final output:

$$\hat{\boldsymbol{V}} = \mathrm{layernorm}_1(\hat{\boldsymbol{V}}_1) + \mathrm{layernorm}_2(\hat{\boldsymbol{V}}_2). \tag{2}$$

The usage of layer norm is to balance the output scales of column and row self-attentions. A similar trick has been used in (Zhu et al., 2021), where the layer norm is applied to resolve scale mismatches between the different attention mechanisms.

Before going to the detailed analysis, we first introduce incoherence parameter of a matrix, which is commonly used in many low-rank matrix applications.

**Definition 1** ($\mu$-incoherence). *Given a rank-$r$ matrix $\boldsymbol{X} \in \mathbb{R}^{n \times d}$. Let $\boldsymbol{X} = \boldsymbol{W} \boldsymbol{\Sigma} \boldsymbol{V}^\top$ be its compact singular value decomposition. $\boldsymbol{X}$ is $\mu$-incoherent if there exists a constant $\mu$ such that*

$$\max_i \|\boldsymbol{e}_i^\top \boldsymbol{W}\| \le \sqrt{\frac{\mu r}{n}} \qquad \text{and} \qquad \max_i \|\boldsymbol{e}_i^\top \boldsymbol{V}\| \le \sqrt{\frac{\mu r}{d}},$$

*where $\boldsymbol{e}_i$ denotes the $i$-th canonical basis vector.*

The $\mu$-incoherence describes the correlation between the column/row spaces and the canonical basis vectors. The larger $\mu$ value implies a higher *overlapping*, which leads to a better chance of successful reconstruction from sparse row/column samples. We next use the following proposition to characterize the efficiency of sampling in both columns and rows.

**Proposition 1.** *Let $\boldsymbol{X} \in \mathbb{R}^{n \times d}$ be a rank-$r$ matrix with $\mu$-incoherence. Without loss of generality, we assume $n \ge d$. Let $\boldsymbol{E} \in \mathbb{R}^{n \times d}$ be a noise matrix. By uniformly sampling $\mathcal{O}(\mu r \log n)$ columns and rows from the noisy $\boldsymbol{X} + \boldsymbol{E}$, Skeleton approximation can construct a matrix $\hat{\boldsymbol{X}}$ such that, with probability at least $1 - \mathcal{O}(n^{-2})$,*

$$\|\boldsymbol{X} - \hat{\boldsymbol{X}}\| \le \mathcal{O}\left( \frac{\|\boldsymbol{E}\|\sqrt{nd}}{\mu r \log n} \right). \tag{3}$$

Several works (e.g., (Chiu & Demanet, 2013; Drineas et al., 2008)) have proposed explicit methods to construct $\hat{\boldsymbol{X}}$. Those methods require computing the pseudo-inverse, generally inefficient in deep learning settings. (Xiong et al., 2021) uses an approximation of the pseudo-inverse in the symmetric matrix setting. It is still an open question whether the approximated pseudo-inverse also works for the general matrix in deep learning settings. On the other hand, in the transformer model, a good matrix approximation is not our primary goal, and we thus pursue a different way that only maintains sufficient information to pass through the network via (2).

## 3.2 SMOOTHER COMPONENT

Based on the analysis of Skeleton approximation, the matrix incoherence parameter $\mu$ plays a crucial role in determining the number of rows and columns to sample. Decreasing in $\mu$ leads to a smaller sampling size. Furthermore, the $\mu$-incoherence condition implies that the "energy" of the matrix is evenly distributed over its entries, i.e., the matrix is "smooth" (Candès & Recht, 2009). In this subsection, we propose a novel smoother component to reduce the incoherence parameter without introducing excessive information loss.

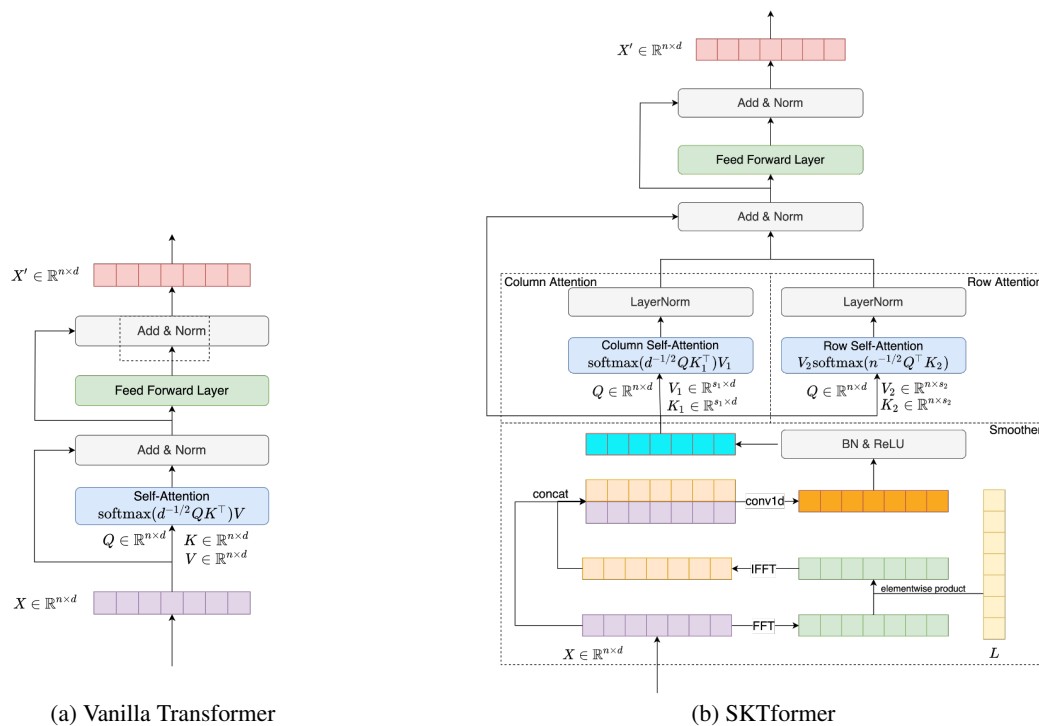

(a) Vanilla Transformer           (b) SKTformer

Figure 1: Illustration of the architecture of Vanilla Transformer versus SKTformer

### 3.2.1 FOURIER CONVOLUTION

The incoherence parameter can be viewed as a measure of the smoothness of a matrix. A "smoother" matrix tends to have a smaller incoherence parameter. Intuitively, the adjacent columns or rows have similar values for a smooth matrix. Thus a few landmark columns or rows can represent the matrix with little error. On the other hand, if the matrix is harsh (e.g., containing spiky columns or rows), more landmarks are required. A common way to smooth a matrix is to convolute it with a smoothing kernel, such as a Gaussian kernel. However, directly using a fixed smoothing kernel can potentially remove too much details and harm the final performance. In the recent literature (e.g., Guo et al. 2022), large convolution kernel-based attentions show a supreme performance in vision Transformers. In this paper, we propose to use a data-driven convolution layer along the sequence dimension with a kernel size equal to the sequence length. In this setting, the information of a given row could be decentralized among the rows. As the input token matrix is computed through a FeedForward layer, the information among different rows is already adaptive allocated. Hence, we do not perform the convolution along the hidden dimension.

We use the Fast Fourier Transformation (FFT) to implement the convolution. Let $\boldsymbol{L}_0 \in \mathbb{R}^{n \times d}$ be the convolution kernel matrix. Via the convolution theorem, the circular convolutions in the spatial domain are equivalent to pointwise products in the Fourier domain, and we then have:

$$\boldsymbol{X}^{\text{smooth}} = \boldsymbol{X} * \boldsymbol{L}_0 = \mathcal{F}^{-1}\left[\mathcal{F}(\boldsymbol{X}) \cdot \mathcal{F}(\boldsymbol{L}_0)\right], \tag{4}$$

where $\mathcal{F}$, $*$, and $\cdot$ denote FFT operator, convolution operator, and point-wise product, respectively.

Equation (4) requires $3d$ times faster Fourier operations which could be prohibited when facing large $d$. In order to save the computational cost, we use the learnable matrix $\boldsymbol{L} \in \mathbb{C}^{n \times d1}$ in the frequency domain instead and apply segment-average (averaging segments of hidden dimension) to $\boldsymbol{X}$. To simplify the notation, we assume there are integers $s$ and $r$ with $d = sr$. Instead of using (4), we apply the following (5) to smooth the token matrix.

$$\boldsymbol{X}^{\text{smooth}} = \mathcal{F}^{-1}\left[\mathcal{F}(\boldsymbol{X}\boldsymbol{S}) \cdot \boldsymbol{L}\right], \tag{5}$$

---

[1]In practice, we use the rFFT/irFFT, the fast (inverse) Fourier Transformation of real input instead of the general FFT/IFFT, and the size of the matrix $\boldsymbol{L}$ is reduced to $\boldsymbol{L} \in \mathbb{C}^{\lfloor n/2 \rfloor + 1) \times d}$.

where

$$\boldsymbol{S} = \begin{bmatrix} \frac{1}{s}\mathbf{1} & \mathbf{0} & ... & \mathbf{0} \\ \mathbf{0} & \frac{1}{s}\mathbf{1} & ... & \mathbf{0} \\ \vdots & \vdots & \ddots & \vdots \\ \mathbf{0} & \mathbf{0} & \cdots & \frac{1}{s}\mathbf{1} \end{bmatrix} \in \mathbb{R}^{d \times d} \tag{6}$$

and $\mathbf{1}$ denotes the $s \times s$ matrix with all elements equal 1. As $\boldsymbol{XS}$ contains repeated rows, in (5), we can reduce the usage of faster Fourier operations to $r + d$ times.

In the following proposition, we show the smooth ability of the Fourier convolution.

**Proposition 2.** *Let* $\{x_1, ...., x_n\}$ *be a sequence with* $\max_t |x_t| \le a_{\max}$ *and* $\max_t |x_t - x_{t-1}| \le b_{\max}$. *Let* $\{l_1, ..., l_n\}$ *be a sequence of i.i.d.* $\frac{1}{n^2}\sigma^2$-*subgaussian variables. Let* $f(t)$ *be the convolution of* $\{x_t\}$ *and* $\{l_t\}$, *i.e.,* $f(t) = \sum_{i=1}^{t} l_{t+1-i}x_i$. *With probability at least* $1 - \delta$, *we have:*

$$|f(t) - f(t-1)| \le b_{\max}\sigma\sqrt{\frac{1}{2n}\log\left(\frac{2n}{\delta}\right)} + a_{\max}\sigma\sqrt{\frac{1}{2n^2}\log\left(\frac{2}{\delta}\right)}. \tag{7}$$

The Proposition 2 can be used to describe the Fourier convolution layer's behavior in the early training stage. Via some standard initialization methods (e.g., Kaiming initialization or Xavier initialization), the variance of elements in learnable matrix $\boldsymbol{L}$ is $\mathcal{O}(n^{-1})$ and the scale of elements is $\mathcal{O}(n^{-1/2})$.[2] To simplify our discussion, let us assume we use Kaiming normal initialization and $\boldsymbol{L}$ becomes a random complex Gaussian matrix with zero mean and variance $n^{-1}\sigma^2$. Using the fact that the FFT of a Gaussian sequence remains Gaussian with $2n$ times larger variance, the $n^{-1}\sigma^2$ variance Gaussian sequence through inverse FFT (IFFT) would result in a Gaussian sequence with $\frac{1}{2n^2}\sigma^2$ variance. By Proposition 2, the maximum difference between adjacent elements after the convolution is scaled on $b_{\max}\sigma n^{-1/2} + a_{\max}\sigma n^{-1} \approx b_{\max}\sigma n^{-1/2}$ when sequence length $n$ is large enough. Thus as long as $\sigma < \mathcal{O}(\sqrt{n})$, the sequence is smoothed by the Fourier convolution.

During the training process, the elements in learnable matrix $\boldsymbol{L}$ go away from the independent random variables and help generate a better representation of segment-averaged token matrix $\boldsymbol{XS}$. We use the following Proposition 3 to describe the potential representation ability of the proposed Fourier convolution component.

**Proposition 3.** *Let* $\boldsymbol{X} \in \mathbb{R}^{n \times d}$ *be a bounded matrix and* $\boldsymbol{S} \in \mathbb{R}^{d \times d}$ *constructed by* (6). *There exist matrices* $\boldsymbol{G}, \boldsymbol{L} \in \mathbb{R}^{n \times d}$ *such that*

$$\left\|(\boldsymbol{XS})_{1:t} - \boldsymbol{X}_t^{\mathrm{smooth}}\boldsymbol{G}_{1:t}\right\| \le \mathcal{O}\left(r^{3/2}t\log(n)d^{-1/2}\right), \tag{8}$$

*where* $(\cdot)_{1:t}$ *is the submatrix of the first* $t$ *rows of a given matrix,* $\boldsymbol{X}_t^{\mathrm{smooth}}$ *is the* $t$-*th row of* $\boldsymbol{X}^{\mathrm{smooth}} = \mathcal{F}^{-1}\left[\mathcal{F}(\boldsymbol{XS}) \cdot \boldsymbol{L}\right]$, *and* $\boldsymbol{G}$ *satisfies* $\boldsymbol{G}_{i,j} = \boldsymbol{G}_{i+s,j} = .... = \boldsymbol{G}_{i+r(s-1),j} = g_i(j)$. *Here* $\{g_1(\cdot), ..., g_s(\cdot)\}$ *is an orthogonal polynomial basis.*

The Proposition 3 states that if we properly train the matrix $\boldsymbol{L}$, the information in $\boldsymbol{XS}$ up to row $t$ can be compressed into $t$-th row of $\boldsymbol{X}^{\mathrm{smooth}}$ with a moderate tolerance. Therefore, when we sample in rows $\boldsymbol{X}^{\mathrm{smooth}}$, they will contain more information than the same number of rows in the original $\boldsymbol{XS}$. Similar results are also discussed in FNet (Lee-Thorp et al., 2021) and several RRN literature, such as (Gu et al., 2020) and (Voelker et al., 2019). In (Gu et al., 2020), several specific types of polynomials (e.g., Legendre or Chebyshev) are explored, and the corresponding matrix $\boldsymbol{L}$ is predefined instead of data-driven. Recently, (Gu et al., 2021b) propose a sophisticated method that can be used to compute $\boldsymbol{X}^{\mathrm{smooth}}$. We leave it for future work.

### 3.2.2 CONVOLUTION STEM

The $\boldsymbol{X}^{\mathrm{smooth}}$ may encounter an over-smoothing situation that local details can be wiped out. We use a convolution stem (CNNs + BN + ReLU) to tackle this problem. We first concatenate $\boldsymbol{X}^{\mathrm{smooth}}$ with the original token matrix $\boldsymbol{X}$ into a $n \times 2d$ matrix and then apply a 1D convolution with kernel size 3 to transform it back to $n \times d$ dimensions. At last, the output is normalized with the Batchnorm layer and truncated by the ReLU activation function to stabilize the training procedure. (Wang et al., 2021) report the ReLU activation coupled with the normalization layer plays an important role in various vision transformers and analyzes this phenomenon theoretically.

---

[2]Here we omit the dependence in $d$ for brevity.

## 4 EXPERIMENTS

In this section, we test our SKTformer on Long Range Arena (LRA) datasets (Tay et al., 2021) and six real-world time series benchmark dataset for long term forecasting. We also evaluate the transfer learning ability of SKTformer on GLUE tasks. We implement the SKTformer based on the official codes of (Zhu et al., 2021) and (Zhou et al., 2022) for LRA and time-series forecasting tasks respectively. The implementation detail (source code) for SKTformer is provided in Appendix A.

### 4.1 LONG-RANGE ARENA

The open-source Long-Range Arena (LRA) benchmark (Tay et al., 2021) is proposed as a standard way to test the capabilities of transformer variants architectures on long sequence tasks.

Table 1: Experimental results on Long-Range Arena benchmark. Best model is in boldface and second best is underlined. The standard deviation of the SKTformer are reported in parenthesis.

| Model | ListOps | Text | Retrieval | Image | Pathfinder | Average |
|---|---|---|---|---|---|---|
| Transformer | 15.82 | 52.98 | 53.39 | 41.46 | 66.63 | 46.06 |
| Local Attention | 15.82 | 52.98 | 53.39 | 41.46 | 66.63 | 46.06 |
| Sparse Transformer | 17.07 | 63.58 | 59.59 | 44.24 | 71.71 | 51.24 |
| Longformer | 35.63 | 62.85 | 56.89 | 42.22 | 69.71 | 53.46 |
| Linformer | 35.70 | 53.94 | 52.27 | 38.56 | 76.34 | 51.36 |
| Reformer | 37.27 | 56.10 | 53.40 | 38.07 | 68.50 | 50.67 |
| Sinkhorn Transformer | 33.67 | 61.20 | 53.83 | 41.23 | 67.45 | 51.39 |
| Synthesizer | 36.99 | 61.68 | 54.67 | 41.61 | 69.45 | 52.88 |
| BigBird | 36.05 | 64.02 | 59.29 | 40.83 | 74.87 | 55.01 |
| Linear Transformer | 16.13 | 65.90 | 53.09 | 42.34 | 75.30 | 50.55 |
| Performer | 18.01 | 65.40 | 53.82 | 42.77 | 77.05 | 51.41 |
| Nystromformer | 37.34 | 65.75 | 81.29 | 41.58 | 70.94 | 59.38 |
| H-Transformer-1D | **49.53** | **78.69** | 63.99 | 46.05 | 68.78 | 61.41 |
| Transformer-LS | 38.36 | 68.40 | 81.85 | 45.05 | 76.48 | 62.03 |
| FNet | 35.33 | 65.11 | 59.61 | 38.67 | 77.08 | 54.42 |
| Luna | 38.01 | 65.78 | 79.56 | 47.86 | **78.89** | 62.02 |
| FMMformer | 36.74 | 67.84 | 81.88 | 45.10 | 72.12 | 60.74 |
| PoNet | 38.80 | 69.82 | 80.35 | 46.88 | 70.39 | 61.05 |
| Cosformer | 37.9 | 63.41 | 61.36 | 43.17 | 70.33 | 55.23 |
| Scatterbrain | 38.6 | 64.55 | 80.22 | 43.65 | 69.91 | 59.38 |
| SKTformer $(r, s_1, s_2 = 8)$ | 38.30(0.40) | 69.27(0.83) | 83.26(0.45) | 53.90(1.54) | 75.82(0.97) | 64.11(2.07) |
| SKTformer (best) | 39.15(0.48) | 71.58(0.95) | **83.73(0.61)** | **57.73(1.83)** | 78.20(1.32) | **66.08(2.56)** |

We benchmark our model with several recent state-of-art efficient transformers, including Sparse Transformer, Longformer , Linformer, Reformers, Sinkhorn Transformer, Synthesizer, BigBird, Linear Transformers, Performer , Nyströmformer , H-Transformer-1D, Transformer-LS, FNet, Luna, FMMformer, Cosformer and Scatterbrain. SKTformer achieves the highest 66.08% average accuracy with tuned parameters and second best 64.11% result with fixed parameters as shown in Table 1.

In particular, SKTformer significantly outperforms the benchmarks on Image tasks by relatively large margins (12.6% and 20.6%, respectively), which support SKTformer's smoothness effect on the low-level features and will benefit the high-level image classification tasks.

Moreover, we want to highlight the sampling efficiency of SKTformer. The sequence length of LRA tasks is over one thousand. The efficient Transformers in literature usually can not project the token matrix to a very small size while maintaining comparable numerical performance, by only sampling 8 rows and columns from the token matrix, SKTformer has already obtained 64.11% average score improving the previous best 62.03% score of Transformer-LS.

### 4.2 LONG-TERM FORECASTING TASKS FOR TIME SERIES

To further evaluate the proposed SKTformer, we also conduct extensive experiments on six popular real-world benchmark datasets for long-term time series forecasting, including traffic, energy, economics, weather, and disease as shown in table 2

To highlight the relevant comparison, we mainly include five state-of-the-art (SOTA) Transformer-based models, i.e., FEDformer(Zhou et al., 2022), Autoformer (Wu et al., 2021), Informer (Zhou et al., 2021a), LogTrans (Li et al., 2019), Reformer (Kitaev et al., 2020a), and one recent state-space

model with recursive memory S4 (Gu et al., 2021a), for comparison. FEDformer is selected as the main baseline as it achieves SOTA results in most settings. More details about baseline models, datasets, and implementations are described in Appendix.

Compared with SOTA work (FEDformer), our proposed SKTformer yields a comparable performance in those tasks, with 4/6 datasets having relative MSE reductions. It is worth noting that the improvement is even more significant on certain datasets, e.g., Exchange ($> 30\%$). Although Exchange does not exhibit an apparent periodicity pattern, SKTformer still achieves superior performance.

Table 2: multivariate long-term series forecasting results on six datasets with input length of 96 and prediction length $O \in \{96, 192, 336, 720\}$ (For ILI dataset, we set prediction length $O \in \{24, 36, 48, 60\}$) with input length 60. A lower MSE indicates better performance. All experiments are repeated 5 times.

| Methods | | SKTformer | | FEDformer | | Autoformer | | S4 | | Informer | | LogTrans | | Reformer | |
|---|---|---|---|---|---|---|---|---|---|---|---|---|---|---|---|---|
| Metric | | MSE | MAE | MSE | MAE | MSE | MAE | MSE | MAE | MSE | MAE | MSE | MAE | MSE | MAE |
| ETTm2 | 96 | **0.192** | **0.283** | 0.203 | 0.287 | 0.255 | 0.339 | 0.705 | 0.690 | 0.365 | 0.453 | 0.768 | 0.642 | 0.658 | 0.619 |
| | 192 | **0.255** | **0.324** | 0.269 | 0.328 | 0.281 | 0.340 | 0.924 | 0.692 | 0.533 | 0.563 | 0.989 | 0.757 | 1.078 | 0.827 |
| | 336 | **0.324** | **0.364** | 0.325 | 0.366 | 0.339 | 0.372 | 1.364 | 0.877 | 1.363 | 0.887 | 1.334 | 0.872 | 1.549 | 0.972 |
| | 720 | 0.431 | 0.433 | **0.421** | **0.415** | 0.422 | 0.419 | 0.877 | 1.074 | 3.379 | 1.338 | 3.048 | 1.328 | 2.631 | 1.242 |
| Electricity | 96 | 0.218 | 0.332 | **0.183** | **0.297** | 0.201 | 0.317 | 0.304 | 0.405 | 0.274 | 0.368 | 0.258 | 0.357 | 0.312 | 0.402 |
| | 192 | 0.259 | 0.361 | **0.195** | **0.308** | 0.222 | 0.334 | 0.313 | 0.413 | 0.296 | 0.386 | 0.266 | 0.368 | 0.348 | 0.433 |
| | 336 | 0.267 | 0.367 | **0.212** | **0.313** | 0.231 | 0.338 | 0.290 | 0.381 | 0.300 | 0.394 | 0.280 | 0.380 | 0.350 | 0.433 |
| | 720 | 0.293 | 0.385 | **0.231** | **0.343** | 0.254 | 0.361 | 0.262 | 0.344 | 0.373 | 0.439 | 0.283 | 0.376 | 0.340 | 0.420 |
| Exchange | 96 | **0.086** | **0.204** | 0.139 | 0.276 | 0.197 | 0.323 | 1.292 | 0.849 | 0.847 | 0.752 | 0.968 | 0.812 | 1.065 | 0.829 |
| | 192 | **0.188** | **0.292** | 0.256 | 0.369 | 0.300 | 0.369 | 1.631 | 0.968 | 1.204 | 0.895 | 1.040 | 0.851 | 1.188 | 0.906 |
| | 336 | **0.356** | **0.433** | 0.426 | 0.464 | 0.509 | 0.524 | 2.225 | 1.145 | 1.672 | 1.036 | 1.659 | 1.081 | 1.357 | 0.976 |
| | 720 | **0.727** | **0.669** | 1.090 | 0.800 | 1.447 | 0.941 | 2.521 | 1.245 | 2.478 | 1.310 | 1.941 | 1.127 | 1.510 | 1.016 |
| Traffic | 96 | 0.592 | 0.352 | **0.562** | **0.349** | 0.613 | 0.388 | 0.824 | 0.514 | 0.719 | 0.391 | 0.684 | 0.384 | 0.732 | 0.423 |
| | 192 | 0.583 | 0.343 | **0.562** | **0.346** | 0.616 | 0.382 | 1.106 | 0.672 | 0.696 | 0.379 | 0.685 | 0.390 | 0.733 | 0.420 |
| | 336 | 0.598 | 0.346 | **0.570** | **0.323** | 0.622 | 0.337 | 1.084 | 0.627 | 0.777 | 0.420 | 0.733 | 0.408 | 0.742 | 0.420 |
| | 720 | 0.641 | 0.397 | **0.596** | **0.368** | 0.660 | 0.408 | 1.536 | 0.845 | 0.864 | 0.472 | 0.717 | 0.396 | 0.755 | 0.423 |
| Weather | 96 | **0.182** | **0.262** | 0.217 | 0.296 | 0.266 | 0.336 | 0.406 | 0.444 | 0.300 | 0.384 | 0.458 | 0.490 | 0.689 | 0.596 |
| | 192 | **0.228** | **0.306** | 0.276 | 0.336 | 0.307 | 0.367 | 0.525 | 0.527 | 0.598 | 0.544 | 0.658 | 0.589 | 0.752 | 0.638 |
| | 336 | **0.295** | **0.355** | 0.339 | 0.380 | 0.359 | 0.395 | 0.531 | 0.539 | 0.578 | 0.523 | 0.797 | 0.652 | 0.639 | 0.596 |
| | 720 | **0.383** | **0.418** | 0.403 | 0.428 | 0.578 | 0.578 | 0.419 | 0.428 | 1.059 | 0.741 | 0.869 | 0.675 | 1.130 | 0.792 |
| ILI | 24 | **2.185** | **0.926** | 2.203 | 0.963 | 3.483 | 1.287 | 4.631 | 1.484 | 5.764 | 1.677 | 4.480 | 1.444 | 4.400 | 1.382 |
| | 36 | **2.155** | **0.937** | 2.272 | 0.976 | 3.103 | 1.148 | 4.123 | 1.348 | 4.755 | 1.467 | 4.799 | 1.467 | 4.783 | 1.448 |
| | 48 | 2.333 | 0.954 | **2.209** | **0.981** | 2.669 | 1.085 | 4.066 | 1.36 | 4.763 | 1.469 | 4.800 | 1.468 | 4.832 | 1.465 |
| | 60 | **2.018** | **0.958** | 2.545 | 1.061 | 2.770 | 1.125 | 4.278 | 1.41 | 5.264 | 1.564 | 5.278 | 1.560 | 4.882 | 1.483 |

## 4.3 TRANSFER LEARNING IN GLUE TASKS

We evaluate the transfer learning ability of the proposed model in the pretraining-finetuning paradigm in NLP tasks. We pretrain vanilla BERT (Devlin et al., 2018), FNet (Lee-Thorp et al., 2021), PoNet (Tan et al., 2021) and our SKTformer with the same MLM loss in (Devlin et al., 2018) on English Wikitext-103 and BooksCorpus datasets. All models are uncased and pretrained with the same configuration with 1 million steps at most. We report the best GLUE results for each model from multiple hyper-parameters configurations in Table 3, and the detailed training configurations in Table 15 in Appendix K. Our SKTformer reaches 77.01 average scores (**96.0%** of the accuracy of vanilla BERT), which also outperform FNet by **4.6%** and PoNet by 0.3% relatively.

Table 3: GLUE validation results. We report the mean of accuracy and F1 for QQP and MRPC, matthew correlations for CoLA, spearman correlations for STS-B, and accuracy for other tasks. For MNLI task, we consider the matched test set.

| Model | MNLI | QQP | QNLI | SST-2 | CoLA | STS-B | MRPC | RTE | Average |
|---|---|---|---|---|---|---|---|---|---|
| BERT-Base | 81.98 | 89.25 | 88.22 | 91.07 | 48.08 | 87.98 | 86.43 | 69.98 | 80.37 |
| FNet-Base | 73.20 | 85.83 | 80.57 | 88.66 | 40.67 | 80.64 | 80.88 | 57.41 | 73.48 |
| PoNet-Base | 77.02 | 87.59 | 84.37 | 89.29 | 45.38 | 84.66 | 81.82 | 64.27 | 76.80 |
| SKTformer (Ours) | 76.86 | 87.67 | 84.12 | 90.14 | 46.72 | 84.87 | 81.84 | 63.87 | 77.01 |

## 4.4 TRAINING SPEED AND PEAK MEMORY USAGE

We compared the training speed (in terms of steps per second) and peak memory usage with several baseline models. SKTformer achieves a 4x time speed advantage and 87% memory reduction

compared to vanilla transformer models with 3k input setting and has a neck-to-neck performance compared to the most efficient baseline models as shown in Table 4.

Table 4: Benchmark results of all Xformer models with a consistent batch size of 32 across all models with various of input lengths

| Model | Train Speed (Steps per second) | | | | Peak Memory Usage (GB) | | | |
|---|---|---|---|---|---|---|---|---|
| | 1 K | 2 K | 3 K | 4 K | 1 K | 2 K | 3 K | 4 K |
| Transformer | 23.8 | 7.8 | 3.9 | *OOM* | 3.7 | 11.1 | 22.1 | *OOM* |
| Linformer | 37.0(1.5x) | 20.8(2.6x) | 14.9(3.7x) | 11.9 | 2.3 | 3.3 | 4.3 | 5.2 |
| Reformer | 28.5(1.2x) | 15.1(1.9x) | 11.9(3.0x) | 9.1 | 2.2 | 3.2 | 4.2 | 4.9 |
| Nystroformer | 33.3(1.4x) | 22.7(2.9x) | 17.2(4.3x) | 14.7 | 1.6 | 2.2 | 2.4 | 2.9 |
| Performer | 29.4(1.2x) | 16.9(1.9x) | 8.7(11.7x) | 9.3 | 2.3 | 3.1 | 4.0 | 4.8 |
| SKTformer | 32.2(1.4x) | 20.4(2.6x) | 15.9(4.0x) | 12.3 | 1.8 | 2.3 | 2.8 | 3.2 |

### 4.5 ROBUSTNESS ANALYSIS

We conduct a noise-resistant experiment for SKTformer and Xformers as shown in Table 5. We use the Image experiment setting in LRA datasets. During generating a sample sequence, we randomly add noise with uniform distribution $\mathcal{U}(-a, a)$ to each position in the sequence. We consider $a \in [0, 2, 4, 8]$ and train every model with 5k steps and 5 replicates. Our model's performance remains robust with a high level of noise injection. This supports our theoretical robustness analysis and shows SKTformer indeed makes an appropriate tradeoff between information preservation and noise reduction.

### 4.6 ABLATION STUDY

This subsection provides an ablation test on four components: Fourier Convolution, Convolution Stem, Column Attention, and Row Attention. We use SKTformer with $(r, s_1, s_2 = 8)$ as the baseline, and the detailed settings are in Table 11 in Appendix F. In Table 6 we present the accuracy changes when removing each component. The performance-decreasing results in Table 6 indicate the four components used in SKTformer are necessary to reach promising results. The most significant component is Column Attention which leads 8.28 average accuracy difference. It reflects that a good summary of the whole sequence is important. Similar observations are also reported in Transformer-LS (Zhu et al., 2021) and XCiT (Ali et al., 2021), where the spirit of attention over columns is used in the dynamic project and Cross-Covariance Attention, respectively. The second most effective part is Fourier Convolution. It reaches a 13.89% accuracy difference in the Retrieval task involving two 4k sequences. Fourier Convolution also works well on shorter sequence tasks (e.g., Image and Pathfinder) and brings a 6.12% accuracy difference.

## 5 CONCLUDING REMARKS

We propose SKTformer, a robust and efficient transformer architecture for modeling long sequences with a good balance between feature preserving and noise resistance. It aggregates a Fourier convolutional stem smoothing information among tokens and a Skeleton-decomposition-inspired efficient self-attention. In particular, our proposed Skeleton Attention directly samples the columns and rows of the token matrix. Such a design increases the model's robustness and gives us a positive near-linear complexity side effect. We conduct a thorough theoretical and experimental analysis of the proposed model and show its effectiveness. Lastly, extensive experiments show that the proposed model achieves the best performance on Long Range Arena compared to all transformer-based baselines and a state-of-art performance in long-term time series forecasting tasks.

Table 5: Average Accuracy on Image task (CIFAR-10 dataset) in Long Range Arena with noise injections. The relative performance changes are reported in parentheses.

| Noise level | 0 | 2 | 4 | 8 |
|---|---|---|---|---|
| Transformer | 41.39 | 40.29 (-2.82%) | 28.56 (-31.12%) | 28.12 (-32.18%) |
| Linformer | 38.43 | 37.99 (-1.49%) | 37.04 (-3.95%) | 36.65 (-4.97%) |
| Reformer | 38.04 | 37.64 (-1.12%) | 35.26 (-7.37%) | 34.88 (-8.37%) |
| Nystroformer | 41.52 | 40.89 (-1.66%) | 38.39 (-7.67%) | 37.84 (-8.99%) |
| Performer | 42.66 | 41.95 (-1.93%) | 39.61 (-7.40%) | 38.86 (-9.15%) |
| SKTformer | 57.47 | 57.06 (-0.82%) | 55.32 (-3.84%) | 54.70 (-4.92%) |

Table 6: Ablation experiments. The SKT $(r, s_1, s_2 = 8)$ is used as baseline. The differences by removing each component from the baseline model are reported.

| Model | LisOps | Text | Retrieval | Image | Pathfinder | Average |
|-------|--------|------|-----------|-------|------------|---------|
| Baseline | 38.30 | 69.27 | 83.26 | 53.90 | 75.82 | 64.11 |
| Fourier Conv. | -0.47 | -4.04 | -13.98 | -5.64 | -6.59 | -6.14 |
| Conv Stem | -0.13 | -0.55 | -1.51 | -1.76 | -9.47 | -0.88 |
| Column Attn. | -1.16 | -8.00 | -9.16 | -10.63 | -12.45 | -8.28 |
| Row Attn. | -0.38 | -1.92 | -1.97 | -2.64 | -2.56 | -1.89 |

One limitation of the current SKTformer is that we need to use both FFT and IFFT in a sequential manner, which is potentially slower than the existing Fourier-based Transformers (e.g., (Lee-Thorp et al., 2021)) that only involve the FFT. As our primary goal using Fourier convolution is to smooth the token matrix and reduce the incoherent parameter, we can use the Random Fourier Transformation (Ailon & Chazelle, 2006) to modify SKTformer with only FFT. Another limitation is that the size of $L$ matrix in the Fourier Convolution part is the same as the input sequence. On a longer sequence, $L$ will contain more learnable parameters that make the model easier to overfit. We may introduce low-rankness or use a more sophisticated design, such as (Gu et al., 2021b), to tackle this issue in the future.

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

# Supplementary Material for
# SKTformer: An Efficient Skeleton Transformer
# for Long Sequence Data

## A  ALGORITHMS

---

**Algorithm 1** Skeleton Attention

---

```
class Skeleton_Attention(nn.Module):
 def __init__(self, num_head = 2, head_dim = 32,seq_len, left_rank = 8,right_rank = 8,
     dropout = 0.1):
    super(Skeleton_Attention, self).__init__()
    self.num_head = num_head
    self.head_dim = head_dim
    self.seq_len = seq_len
    self.left_rank = left_rank
    self.right_rank = right_rank

    self.ln_1 = nn.LayerNorm(self.num_head * self.head_dim)
    self.ln_2 = nn.LayerNorm(self.num_head * self.head_dim)

    self.drop_attn = torch.nn.Dropout(p=dropout)

    self.index_set_right = torch.randperm(self.head_dim)
    self.index_set_right = self.index_set_right[:self.right_rank]

    self.index_set_left = torch.randperm(self.seq_len)
    self.index_set_left = self.index_set_left[:self.left_rank]

 def combine_heads(self, X):
    X = X.transpose(1, 2)
    X = X.reshape(X.size(0), X.size(1), self.num_head * self.head_dim)
    return X

 def split_heads(self, X):
    X = X.reshape(X.size(0), X.size(1), self.num_head, self.head_dim)
    X = X.transpose(1, 2)
    return X

 def forward(self,Q, K, V):
    #### Row Attention ####
    if self.left_rank <= self.seq_len:
       K1 = K[:,:,self.index_set_left,:]
       V1 = V[:,:,self.index_set_left,:]
    else:
       K1 = K
       V1 = V

    dots = Q @ K1.transpose(-1,-2)
    dots = dots / math.sqrt(self.head_dim)
    attn = nn.functional.softmax(dots,dim=-1)
    attn = self.drop_attn(attn)

    #### Column Attention ####
    Q2 = Q.transpose(-1,-2)
    if self.right_rank <= self.head_dim:

       K2 = K[:,:,:,self.index_set_right]
       V2 = V[:,:,:,self.index_set_right]
    else:
       K2 = K
       V2 = V

    dots_r = Q2 @ K2
    dots_r = dots_r / math.sqrt(self.seq_len)
    attn_r = nn.functional.softmax(dots_r,dim=-1).transpose(-1,-2)
    attn_r = self.drop_attn(attn_r)

    X = self.split_heads(self.ln_1(self.combine_heads(torch.matmul(attn,V1))))/2 + self.
        split_heads(self.ln_2(self.combine_heads(torch.matmul(V2,attn_r))))/2

    return X
```

---

---

**Algorithm 2** Smoother component

---

```
class Smoother(nn.Module):

    def __init__(self, hidden_size, seq_len, dropout = 0.5, num_head = 2,transformer_dim =
        64, fold = 1):

        super(Smoother, self).__init__()

        self.hidden_size = hidden_size
        self.seq_len = seq_len
        self.dropout = dropout
        self.num_head = num_head
        self.dim = transformer_dim
        self.fold = fold

        self.weights_fft = nn.Parameter(torch.empty(self.seq_len//2+1, self.hidden_size,2))
        nn.init.kaiming_normal_(self.weights_fft, mode='fan_in', nonlinearity='relu')

        self.tiny_conv_linear = torch.nn.Conv1d(in_channels = self.hidden_size*2 ,
            out_channels = self.hidden_size, kernel_size = 3, padding= 1, groups = 1)
        self.dropout = torch.nn.Dropout(p=self.dropout)
        self.bn_1 = nn.BatchNorm1d(self.seq_len)

    def forward(self, x):

        #### Compute Segment Average ####
        B,S,H = x.shape
        u = x.reshape(B,S,self.fold,H//self.fold)
        u = torch.mean(u,dim = -1)

        #### Fourier Convolution ####
        fft_u = fft.rfft(u, n = self.seq_len, axis = -2)
        fft_u = torch.view_as_real(fft_u)
        fft_u = fft_u.repeat(1,1,H//self.fold,1)
        self.weight_used = self.weights_fft.unsqueeze(0)
        temp_real = fft_u[...,0]*self.weight_used[...,0] - fft_u[...,1]*self.weight_used
            [...,1]
        temp_imag = fft_u[...,0]*self.weight_used[...,1] + fft_u[...,1]*self.weight_used
            [...,0]
        out_ft = torch.cat([temp_real.unsqueeze(-1),temp_imag.unsqueeze(-1)],dim = -1)
        out_ft = torch.view_as_complex(out_ft)
        m = fft.irfft(out_ft, n = self.seq_len, axis = -2)

        #### Convolution Stem ####
        input_h = torch.cat((m, x), dim = -1)
        h = self.tiny_conv_linear(input_h.permute(0,2,1)).permute(0,2,1)
        h = self.dropout(F.relu(self.bn_1(h)))

        return h
```

---

**Algorithm 3** pseudo code for Time-Series Forecasting

---

```
    def forward(self, x_in):
        B1,H1,C1 = x_in.shape
        for i in range(len(self.encoder)):
            attn_layer = self.encoder[i]
            #standardize the input data
            if i == 0:
                tmp_mean = torch.mean(x_in[:,:,:],dim = 1,keepdim = True)
                tmp_std = torch.sqrt(torch.var(x_in[:,:,:],dim = 1,keepdim = True)+1e0)
                x_in = (x_in - tmp_mean)/(tmp_std)

                enc_out1 = self.enc_embedding(x_in)

            enc_out1= attn_layer(enc_out1) + enc_out1

        #decoder via Fourier Extrapolation
        dec_out = self.fourierExtrapolation(post(enc_out1))
        output = (dec_out.reshape(B1,-1,C1))*(tmp_std)+tmp_mean
        return output
```

---

# B  PROOF OF PROPOSITION 1

A similar result, under a slightly different setting, can be found in (Cai et al., 2021). For the completeness of the paper, we provide a proof here. We resolve the sampling strategy. We consider a clear rank-$r$ matrix $\boldsymbol{X} \in \mathbb{R}^{n \times d}$, i.e., no additive noise and the rank is exact. Without loss of generality, we assume $n \geq d$. Provided $\boldsymbol{X}$ is $\mu$-incoherent, by (Chiu & Demanet, 2013, Theorem 1.1), Skeleton

---

**Algorithm 4** Fourier Extrapolation

---

```
class fourierExtrapolation(nn.Module):
    def __init__(self,inputSize,n_harm = 8,n_predict = 96):
        super().__init__()
        self.n = inputSize
        self.n_harm = n_harm
        self.f = torch.fft.fftfreq(self.n)
        self.indexes = list(range(self.n))

        # sort indexes by frequency, lower -> higher
        self.indexes.sort(key = lambda i: torch.absolute(self.f[i]))
        self.indexes = self.indexes[:1 + self.n_harm * 2]

        self.n_predict = n_predict

        # compute init phase
        self.t = torch.arange(0, self.n + self.n_predict)
        self.t1 = self.t.unsqueeze(0).unsqueeze(-1).float().to('cuda')
        self.f = self.f.unsqueeze(0).unsqueeze(-1).to('cuda')
        self.t = self.t.unsqueeze(0).unsqueeze(-1).unsqueeze(-1)to('cuda')
        self.g = self.f[:,self.indexes,:].permute(0,2,1).unsqueeze(1)
        self.phase_init = 2 * 3.1415 * self.g * self.t

    def fourierExtrapolation(self,x):
        # x in frequency domain
        x_freqdom = torch.fft.fft(x,dim = -2)
        x_freqdom = torch.view_as_real(x_freqdom)
        # select importance frequencies
        x_freqdom = x_freqdom[:,self.indexes ,:,:]
        x_freqdom = torch.view_as_complex(x_freqdom)
        ampli = torch.absolute(x_freqdom) / self.n # amplitude
        phase = torch.angle(x_freqdom) # phase

        ampli = ampli.permute(0,2,1).unsqueeze(1)
        phase = phase.permute(0,2,1).unsqueeze(1)

        self.restored_sig = ampli * torch.cos(self.phase_init + phase)

        return torch.sum(self.restored_sig,dim = -1)
```

---

approximation recovers $X$ exactly, i.e.,

$$X = CUR,$$

with probability at least $1 - \mathcal{O}(n^{-2})$ if we uniformly sample $\mathcal{O}(\mu r \log n)$ rows and columns to form the submatrices $C$ and $R$.

Thirdly, we resolve the error bound estimation. For the noisy matrix $X + E$, we directly apply (Hamm & Huang, 2021, Corollary 4.3). Thus, we have

$$\|X - \hat{C}\hat{U}\hat{R}\| \leq \mathcal{O}\left(\sqrt{\frac{nd}{l_C l_R}}\right) \|E\|,$$

where $\hat{C}$ and $\hat{R}$ are sampled from the noisy matrix, $\hat{U}$ is the pseudo-inverse of their intersection, and $l_C$ (resp. $l_R$) is the number of columns (resp. rows) being sampled in $\hat{C}$ (resp. $\hat{R}$).

Note that this error bound assumes good column and row sampling, i.e., the clear submatrices corresponding to $\hat{C}$ and $\hat{R}$ can recover $X$ exactly. Therefore, by combining the above two results, we show the claim in Proposition 1.

## C   PROOF OF PROPOSITION 2

As $f(t)$ is the convolution function of $\{x_t\}$ and $\{l_t\}$, from the definition of convolution for $t = 1, 2, ..., n$ we have

$$f(t) = \sum_{i=1}^{t} l_{t+1-i} x_i$$

and

$$f(t) - f(t-1) = \underbrace{\sum_{i=1}^{t-1}(l_{i+1} - l_i)x_i}_{:=(a_t)} + l_1 x_t. \tag{9}$$

By Hoffelding inequality, term $(a)$ satisfies the following inequality with $\varepsilon > 0$.

$$\mathbb{P}(|(a_t)| \geq \varepsilon) = \mathbb{P}\left(\left|\sum_{i=1}^{t-1}(l_{i+1} - l_i)x_i\right| \geq \varepsilon\right) \leq \exp\left(-\frac{2\varepsilon^2}{(t-1)b_{\max}^2 \cdot \frac{1}{n^2}\sigma^2}\right) \tag{10}$$

Combine (10) with the union bound over $t = 1, 2, ..., n$ and the following (10) holds with probability at least $1 - \delta/2$:

$$\max_t |(a_t)| \leq b_{\max}\sigma\sqrt{\frac{1}{2n}\log\left(\frac{2n}{\delta}\right)} \tag{11}$$

Similarly, with probability $1 - \delta/2$, we have

$$\max_t |l_1 x_t| \leq a_{\max}\sigma\sqrt{\frac{1}{2n^2}\log\left(\frac{2}{\delta}\right)}. \tag{12}$$

Therefore, via (11) and (12), with probability at least $1 - \delta$, we have

$$\max_t |f(t) - f(t-1)| \leq b_{\max}\sigma\sqrt{\frac{1}{2n}\log\left(\frac{2n}{\delta}\right)} + a_{\max}\sigma\sqrt{\frac{1}{2n^2}\log\left(\frac{2}{\delta}\right)} \tag{13}$$

## D    PROOF OF PROPOSITION 3

The proof contains two parts. In the first part, we view the data sequence as a function of index $t$ and construct the coefficients and orthogonal polynomials for function approximation. In the second part, we show such coefficients can be computed with Fourier convolution i.e. (5)).

**Function Approximation.**    We reformulate the matrix $\boldsymbol{XS}$ as follow:

$$\boldsymbol{XS} = \begin{bmatrix} \bar{\boldsymbol{x}}_1\boldsymbol{e} & \bar{\boldsymbol{x}}_2\boldsymbol{e} & \cdots & \bar{\boldsymbol{x}}_r\boldsymbol{e} \end{bmatrix},$$

where $\boldsymbol{e} \in \mathbb{R}^{1\times s}$ is the one vector and $\bar{\boldsymbol{x}}_i \in \mathbb{R}^{n\times 1}$ is the average from $(s(i-1)+1)$-th column to $(si)$-th column of $\boldsymbol{X}$.

Next, we focus on vector $\bar{\boldsymbol{x}}_j$ and view its $t$-th element as the output of a function $h^j(t) = \bar{\boldsymbol{x}}_{jt}$. Via analysis in (Gu et al., 2020, Appendixes C and D), we can form an approximation on $h^j(t)$ as follow:

$$h^j_{[x\leq t]}(x) \approx \sum_{i=1} c_i^j(t)g_i(x), \tag{14}$$

where $\{g_i\}$ is a sequence of orthogonal polynomial and $[c_1^j(t), c_2^j(t), ..., c_s^j(t)] := \boldsymbol{c}_t^j \in \mathbb{R}^{1\times s}$ satisfy

$$\frac{d}{dt}\boldsymbol{c}(t)^j = \frac{1}{t}\boldsymbol{c}(t)^j\boldsymbol{A}_0 + \frac{1}{ts\log n}h(t)\boldsymbol{b}_0 \tag{15}$$

where $\boldsymbol{A}_0 \in \mathbb{R}^{s\times s}$ and $\boldsymbol{b}_0 \in \mathbb{R}^{1\times s}$ are predefined matrix and vector respectively. Equation (15) is corresponding to the case with $\lambda_n = s\log n$ in (Gu et al., 2020).

We then use Forward Euler approach to discretize it:

$$\hat{\boldsymbol{c}}(t)^j = \hat{\boldsymbol{c}}(t-1)^j(\frac{1}{t}\boldsymbol{I} + \frac{1}{t}\boldsymbol{A}_0) + \frac{1}{ts\log n}h(t)\boldsymbol{b}_0,. \tag{16}$$

Via standard error analysis of Forward Euler approach, we have

$$
\begin{aligned}
\boldsymbol{c}(t+1)^j &= \boldsymbol{c}(t)^j + \frac{1}{t}\boldsymbol{c}(t)^j \boldsymbol{A}_0 + \frac{1}{ts\log n}h(t)\boldsymbol{b}_0 + \frac{d^2}{dt^2}\boldsymbol{c}(t)^j|_{t=\xi} \\
&= \boldsymbol{c}(t)^j + \frac{1}{t}\boldsymbol{c}(t)^j \boldsymbol{A}_0 + \frac{1}{ts\log n}h(t)\boldsymbol{b}_0 + \frac{1}{\xi s\log n}h(\xi)'\boldsymbol{b}_0 \\
&= \boldsymbol{c}(t)^j + \frac{1}{t}\boldsymbol{c}(t)^j \boldsymbol{A}_0 + \frac{1}{ts\log n}h(t)\boldsymbol{b}_0 + \mathcal{O}\left(\frac{1}{ts\log n}\right),
\end{aligned}
$$

where $\xi \in [t, t+1]$.

It implies that for $t = 1, 2, ..., n$,

$$
\|\hat{\boldsymbol{c}}(t)^j - \boldsymbol{c}(t)^j\| \leq \mathcal{O}\left(\frac{\log t}{s\log n}\right). \tag{17}
$$

Combine (17) with the similar proof procedure in (Gu et al., 2020, Proposition 6), if $h^j(x)$ is quadratic spline interpolation on $\{\bar{\boldsymbol{x}}_{jt}\}$, we obtain

$$
\|\bar{\boldsymbol{x}}_{jt} - \sum_{i=1}^{s}\hat{\boldsymbol{c}}_i(t)g_i(x)\| \leq \mathcal{O}\left(t\log n/\sqrt{s}\right) = \mathcal{O}\left(t\log n\sqrt{\frac{r}{d}}\right). \tag{18}
$$

The desirable result in Proposition 3 is obtained by repeatedly using (18) with $j = 1, 2, ..., r$.

**Coefficients via Fourier Convolution.** The remaining task is to show that $\{\hat{\boldsymbol{c}}(t)^j\}$ can be generated via Fourier convolution. To simplify the notation, we denote $\boldsymbol{A} = \frac{1}{t}\boldsymbol{I} + \frac{1}{t}\boldsymbol{A}_0$ and $\boldsymbol{b} = \frac{1}{t\log n}\boldsymbol{b}_0$ and (16) becomes

$$
\hat{\boldsymbol{c}}(t)^j = \hat{\boldsymbol{c}}(t-1)^j\boldsymbol{A} + h(t)\boldsymbol{b}. \tag{19}
$$

We then repeatedly use (19) from $t = 1, 2, ...$ and one may verify

$$
\begin{aligned}
\hat{\boldsymbol{c}}_t^j &= \sum_{i=1}^{t-1}\boldsymbol{b}\boldsymbol{A}^{t-i}h(i) = \sum_{i=1}^{t-1}\boldsymbol{b}\boldsymbol{A}^{t-i}\bar{\boldsymbol{x}}_{ji} \\
\Rightarrow \quad \boldsymbol{C}^j &= \bar{\boldsymbol{A}}_j * (\bar{\boldsymbol{x}}_j\boldsymbol{e}),
\end{aligned} \tag{20}
$$

where

$$
\boldsymbol{C}^j = \begin{bmatrix} \hat{\boldsymbol{c}}_1^j \\ \hat{\boldsymbol{c}}_2^j \\ \vdots \\ \hat{\boldsymbol{c}}_n^j \end{bmatrix} \in \mathbb{R}^{n\times s}, \quad \text{and} \quad \bar{\boldsymbol{A}}_j = \begin{bmatrix} \boldsymbol{b} \\ \boldsymbol{b}\boldsymbol{A} \\ \vdots \\ \boldsymbol{b}\boldsymbol{A}^{n-1} \end{bmatrix} \in \mathbb{R}^{n\times s}. \tag{21}
$$

Next we repeatedly use (20) from $j = 1, 2, .., r$, and one has

$$
\begin{aligned}
\underbrace{\begin{bmatrix} \boldsymbol{C}^1 & \boldsymbol{C}^2 & \cdots & \boldsymbol{C}^r \end{bmatrix}}_{:=\boldsymbol{X}^{\mathrm{smooth}}} &= \underbrace{\begin{bmatrix} \bar{\boldsymbol{A}}_1 & \bar{\boldsymbol{A}}_2 & \cdots & \bar{\boldsymbol{A}}_r \end{bmatrix}}_{:=\boldsymbol{L}_0} * \underbrace{\begin{bmatrix} \bar{\boldsymbol{x}}_1\boldsymbol{e} & \bar{\boldsymbol{x}}_2\boldsymbol{e} & \cdots & \bar{\boldsymbol{x}}_r\boldsymbol{e} \end{bmatrix}}_{=\boldsymbol{X}\boldsymbol{S}} \\
\Rightarrow \quad \boldsymbol{X}^{\mathrm{smooth}} &= \boldsymbol{L}_0 * \boldsymbol{X}\boldsymbol{S} \\
\Rightarrow \quad \boldsymbol{X}^{\mathrm{smooth}} &= \mathcal{F}^{-1}\left(\mathcal{F}(\boldsymbol{L}_0)\cdot\mathcal{F}(\boldsymbol{X}\boldsymbol{S})\right) \\
\Rightarrow \quad \boldsymbol{X}^{\mathrm{smooth}} &= \mathcal{F}^{-1}\left(\boldsymbol{L}\cdot\mathcal{F}(\boldsymbol{X}\boldsymbol{S})\right),
\end{aligned}
$$

where we use the fact that $\boldsymbol{L}$ is constructed in frequency domain in Fourier convolution in Eq. (5).

# E  MODEL PARAMETERS IMPACT

SKTformer introduces three extra hyperparameters, $r$, $s_1$ and $s_2$. We test the influence when varying them and report results in Table 7. We use SKTformer $(r, s_1, s_2 = 8)$ as the baseline model and other parameters are reported in Table 10 in Appendix F.

Table 7: Experimental results on varying $r$, $s_1$ and $s_2$. Best result is in boldface and second best is underlined. And Ablation experiments for each components

(a) Experimental results on varying $r$ parameter in smoothing component.

| $r$ | LisOps | Text | Retrieval | Image | Pathfinder | Average |
|---|---|---|---|---|---|---|
| 1 | 37.30 | 65.25 | 78.65 | 51.36 | 71.23 | 60.76 |
| 8 | 38.30 | 69.27 | **83.26** | 53.90 | 75.82 | 64.11 |
| 16 | **38.62** | **70.02** | 83.21 | **54.20** | 76.15 | **64.44** |
| 32 | 38.19 | 69.27 | 82.05 | 53.73 | 75.58 | 63.76 |
| 64 | 37.89 | 69.73 | 81.79 | 51.28 | 75.52 | 63.24 |

(b) Experimental results on varying $s_1$ parameter in Row Attention.

| $s_1$ | LisOps | Text | Retrieval | Image | Pathfinder | Average |
|---|---|---|---|---|---|---|
| 8 | 38.30 | 69.27 | 83.26 | 53.90 | 75.82 | 64.11 |
| 32 | **38.44** | **70.85** | **83.41** | **54.92** | 77.97 | **65.12** |
| 64 | 37.88 | 70.53 | 83.02 | 51.22 | 78.02 | 64.33 |
| 128 | 37.33 | 69.24 | 81.58 | 49.08 | 78.12 | 63.07 |
| 256 | 37.02 | 65.72 | 79.30 | 46.24 | **78.14** | 61.29 |

(c) Experimental results on varying $s_2$ parameter in Column Attention.

| $s_2$ | LisOps | Text | Retrieval | Image | Pathfinder | Average |
|---|---|---|---|---|---|---|
| 1 | 37.32 | 55.28 | 57.37 | 40.97 | 66.25 | 51.44 |
| 4 | 37.82 | 52.05 | 72.58 | 46.74 | 73.17 | 57.47 |
| 8 | **38.30** | 69.27 | 83.26 | 53.90 | 75.82 | 64.11 |
| 16 | 37.77 | **70.24** | **83.42** | **54.11** | 77.92 | **64.73** |
| 32 | 37.62 | 68.32 | 80.11 | 51.66 | **78.18** | 62.98 |

(d) The SKT $(r, s_1, s_2 = 8)$ is used as baseline. The differences by removing each component from the baseline model are reported.

| Model | LisOps | Text | Retrieval | Image | Pathfinder | Average |
|---|---|---|---|---|---|---|
| Baseline | 38.30 | 69.27 | 83.26 | 53.90 | 75.82 | 64.11 |
| Fourier Conv. | -0.47 | -4.04 | -13.98 | -5.64 | -6.59 | -6.14 |
| Conv Stem | -0.13 | -0.55 | -1.51 | -1.76 | -9.47 | -0.88 |
| Column Attn. | -1.16 | -8.00 | -9.16 | -10.63 | -12.45 | -8.28 |
| Row Attn. | -0.38 | -1.92 | -1.97 | -2.64 | -2.56 | -1.89 |

**Influence of $r$ in Fourier Convolution.** The $r$ parameter is used to determine the number of segment-averages to compute in (5). The smaller $r$ leads the matrix with more duplicate columns, and more details information is lost. On the other hand, according to Proposition 3, the larger $r$ would potently decrease the memorization ability and yield a high approximation error. In Table 7a, the best performance is observed when $r = 8$ or $r = 16$. For the case with $r = 1$, the token matrix is smoothed to rank one matrix, and the average accuracy drops 3.55 from the best setting. When the $r$ value goes larger than 16, the accuracy in all experiments slightly decreases. We believe it is due to the over-fitting since the smoothed token matrix contains more flexibility and more irrelevant information training dataset is learned.

**Influence of sample number $s_1$ in Row Attention.** In Row Attention part, we randomly sample $s_1$ from key and value tokens. Table 7b reports that the optimal sampling amounts are different among tasks. In Pathfinder task, the optimal result is associated with $s_1 = 256$, while the best performance of other tasks the reached with $s_1 = 32$. Pathfinder task requires learning extreme long-range dependence (the connectivity between two circles far away from each other). The lack of enough tokens leads to inaccurate long-range dependence estimation and damages the final results. For the tasks like Image or Retrieval, the modest range dependence may already be enough to get promising performance, and we thus could use fewer token samples.

**Influence of sample number $s_2$ in Column Attention.** In Column Attention, $s_2$ columns are selected. The experiment results are shown in Table 7c. When setting $s_2 = 1$, average performance decreases by 13.24%. Similar behavior is also observed in the first row of Table 7a with $r = 1$. The information loss due to lack of rankness limits the final performance. In an average sense, $s_2 = 16$ gives the best result, and further increasing in $s_2$ slightly harms the accuracy in all tasks except Pathfinder.

# F  EXPERIMENT CONFIGURATIONS

In this section, we report the configurations for the experiments in Sections 4.1, 4.2, and 4.3.

Table 8: Experiment Configuration of SKTformer $(r, s_1, s_2 = 8)$.

| Parameters | ListOps | Text | Retrieval | Image | Pathfinder |
|---|---|---|---|---|---|
| Epoch | 5 | 30 | 15 | 60 | 100 |
| Learning Rate | 1e-4 | 1e-4 | 1e-4 | 1e-3 | 1e-4 |
| Weight Decay | 0 | 1e-2 | 1e-2 | 1e-2 | 1e-2 |
| Batch Size | 32 | 32 | 32 | 256 | 256 |
| $r, s_1, s_2$ | 8,8,8 | 8,8,8 | 8,8,8 | 8,8,8 | 8,8,8 |
| dropout in embedding | 0 | 0.5 | 0.1 | 0.1 | 0 |
| dropout in attention | 0 | 0.1 | 0.1 | 0.1 | 0 |
| dropout in smoother | 0 | 0.5 | 0.1 | 0.5 | 0.5 |

Table 9: Experiment Configuration of SKTformer (best).

| Parameters | ListOps | Text | Retrieval | Image | Pathfinder |
|---|---|---|---|---|---|
| Epoch | 10 | 30 | 15 | 60 | 100 |
| Learning Rate | 1e-4 | 1e-4 | 1e-4 | 1e-3 | 1e-4 |
| Weight Decay | 1-2 | 1e-2 | 1e-2 | 1e-2 | 1e-2 |
| Batch Size | 32 | 32 | 32 | 256 | 256 |
| $r, s_1, s_2$ | 8,8,8 | 8,8,8 | 8,32,32 | 8,16,16 | 8,128,32 |
| dropout in embedding | 0 | 0.5 | 0.1 | 0.5 | 0.1 |
| dropout in attention | 0 | 0.1 | 0.1 | 0.1 | 0.1 |
| dropout in smoother | 0 | 0.5 | 0.1 | 0.5 | 0.5 |

Table 10: Experiment Configuration for Model Parameters Impact.

| Parameters | ListOps | Text | Retrieval | Image | Pathfinder |
|---|---|---|---|---|---|
| Epoch | 5 | 30 | 15 | 60 | 100 |
| Learning Rate | 1e-4 | 1e-4 | 1e-4 | 1e-3 | 1e-4 |
| Weight Decay | 0 | 1e-2 | 1e-2 | 1e-2 | 1e-2 |
| Batch Size | 32 | 32 | 32 | 256 | 256 |
| dropout in embedding | 0 | 0.5 | 0.1 | 0.1 | 0 |
| dropout in attention | 0 | 0.1 | 0.1 | 0.1 | 0 |
| dropout in smoother | 0 | 0.5 | 0.1 | 0.5 | 0.5 |

Table 11: Experiment Configuration for Ablation.

| Parameters | ListOps | Text | Retrieval | Image | Pathfinder |
|---|---|---|---|---|---|
| Learning Rate | 1e-4 | 1e-4 | 1e-4 | 1e-3 | 1e-4 |
| Weight Decay | 0 | 1e-2 | 1e-2 | 1e-2 | 1e-2 |
| Batch Size | 32 | 32 | 32 | 256 | 256 |
| $r, s_1, s_2$ | 8,8,8 | 8,8,8 | 8,8,8 | 8,8,8 | 8,8,8 |
| dropout in embedding | 0 | 0.5 | 0.1 | 0.1 | 0 |
| dropout in attention | 0 | 0.1 | 0.1 | 0.1 | 0 |
| dropout in smoother | 0 | 0.5 | 0.1 | 0.5 | 0.5 |

## G  ADDITIONAL RESULTS ON LRA

We have already provided the average of 5 runs with different random seeds in Table 1. Here we also provide the standard deviations for these experiments in Table 12.

## H  DATASET AND IMPLEMENTATION DETAILS

In this subsection, we summarize the details of the datasets used in this paper as follows:

Table 12: Accuracy on Long Range Arena (LRA) with standard errors shown in parenthesis. All results are averages of 5 runs with different random seeds.

| Model | LisOps | Text | Retrieval | Image | Pathfinder |
|---|---|---|---|---|---|
| SKTformer $(r, s_1, s_2 = 8)$ | 38.30 (0.40) | 69.27 (0.83) | 83.26 (0.45) | 53.90 (1.54) | 75.82 (0.97) |
| SKTformer (best) | 39.15 (0.48) | 71.58 (0.95) | 83.73 (0.61) | 57.73 (1.83) | 78.20 (1.32) |

Table 13: Details of time series benchmark datasets.

| DATASET | LENGTH | DIMENSION | FREQUENCY |
|---|---|---|---|
| ETTm2 | 69680 | 8 | 15 MIN |
| EXCHANGE | 7588 | 9 | 1 DAY |
| WEATHER | 52696 | 22 | 10 MIN |
| ELECTRICITY | 26304 | 322 | 1H |
| ILI | 966 | 8 | 7 DAYS |
| TRAFFIC | 17544 | 863 | 1H |

LRA datasets: **ListOps**(2K length mathematical expression task which investigates the parsing ability); **Text** (up to 4K byte/character-level document classification task that tests capacity in character compositionality); **Retrieval** (byte/character-level document matching task, which exams the information compression ability with two 4K length sequence); **Image** (pixel-wise sequence image classification based on the CIFAR-10 dataset); **Pathfinder** (long-range spatial dependency identification task. The input images contain two small points/circles and dash-line paths. The model needs to identify whether two points/circles are connected);The LRA has several desirable advantages that made us focus on it as the evaluation benchmark: **generality** (only requires the encoder part); **simplicity** (data augmentation and pretraining are out of scope); **challenging long inputs** (difficulty enough and room to improve); **diversity aspects** (tasks covering math, language, image, and spatial modeling); and **lightweight** (run with low resource requirement).

Time series datasets:1) ETT (Zhou et al., 2021a) dataset contains two sub-dataset: ETT1 and ETT2, collected from two separated counties. Each of them has two versions of sampling resolutions (15min & 1h). ETT dataset contains multiple time series of electrical loads and one time sequence of oil temperature. 2) Electricity[3] dataset contains the electricity consumption for more than three hundred clients with each column corresponding to one client. 3) Exchange (Lai et al., 2018) dataset contains the current exchange of eight countries. 4) Traffic[4] dataset contains the occupation rate of freeway systems in California, USA. 5) Weather[5] dataset contains 21 meteorological indicators for a range of one year in Germany. 6) Illness[6] dataset contains the influenza-like illness patients in the United States. Table 13 summarizes all the features for the six benchmark datasets. They are all split into the training set, validation set and test set by the ratio of 7:1:2 during modeling.

GLUE datasets: The GLUE benchmark covers various natural language understanding tasks and is widely used in evaluating transfering ability. The tasks can be devided in to two types, single-sentence tasks (SST-2 and CoLA), and sentence-pair tasks (MNLI, QQP,QNLI,STS-B,MRPC,RTE). Following the same settings in (Devlin et al., 2018), we exclude WNLI task.

## I  EXPERIMENTS ON THE SMOOTHNESS EFFECT OF FOURIER CONVOLUTION

In this section, we verify Fourier convolution component in the Smoother block can reduce the incoherence value in the early training stage. We use SKTformer with $(r, s_1, s_2 = 8)$ as the test model and test on an NLP dataset: Text, and a vision dataset: Pathfinder. We compute the $\mu$-incoherence value [7] of the token matrix before and after the Fourier convolution (denoted as $\mu_{\boldsymbol{X}}$

---

[3]https://archive.ics.uci.edu/ml/datasets/ElectricityLoadDiagrams 20112014

[4]http://pems.dot.ca.gov

[5]https://www.bgc-jena.mpg.de/wetter

[6]https://gis.cdc.gov/grasp/fluview/fluportaldashboard.html

[7]Incoherence is defined by Definition 1 in Appendix B.

and $\mu_{\boldsymbol{X}^{\text{smooth}}}$, respectively) for each samples in the validation dataset. Since we do not explicitly force the token matrix to be low-rank required by Definition 1, we report the incoherence value for different rankness settings ($\text{rank} = 16$ and $\text{rank} = 32$) approximately, and the mean and standard deviation of incoherence value can be found in Table 14. The average incoherence value reduced 30% after the Fourier convolution in both datasets. Moreover, We observe that the standard deviation significantly decreases, which suggests the Fourier convolution may also potentially stabilize the training procedure.

Table 14: The average incoherence parameters after 100 training steps with standard errors shown in the parenthesis.

| Dataset | $\mu_{\boldsymbol{X}}$ ($\text{rank} = 32$) | $\mu_{\boldsymbol{X}^{\text{smooth}}}$ ($\text{rank} = 32$) | $\mu_{\boldsymbol{X}}$ ($\text{rank} = 16$) | $\mu_{\boldsymbol{X}^{\text{smooth}}}$ ($\text{rank} = 16$) |
|---|---|---|---|---|
| Text | 2.75 (0.027) | 2.05 (0.007) | 3.98 (0.046) | 3.23 (0.038) |
| Pathfinder | 3.83 (0.221) | 1.99 (0.001) | 4.88 (0.264) | 3.48 (0.001) |

## J    ILLUSTRATION ON EFFECT OF THE SMOOTHER AND SKELETON ATTENTION IN TOKEN MATRIX

In this section, an illustration of the Smoother and Skeleton Attention part is shown in Figure 2. We smooth the input token matrix to ensure the sampling in rows and columns containing more local and/or global information. Thus, sampling several rows and columns from the smoothed token matrix can be more effective than the samples from the original token matrix.

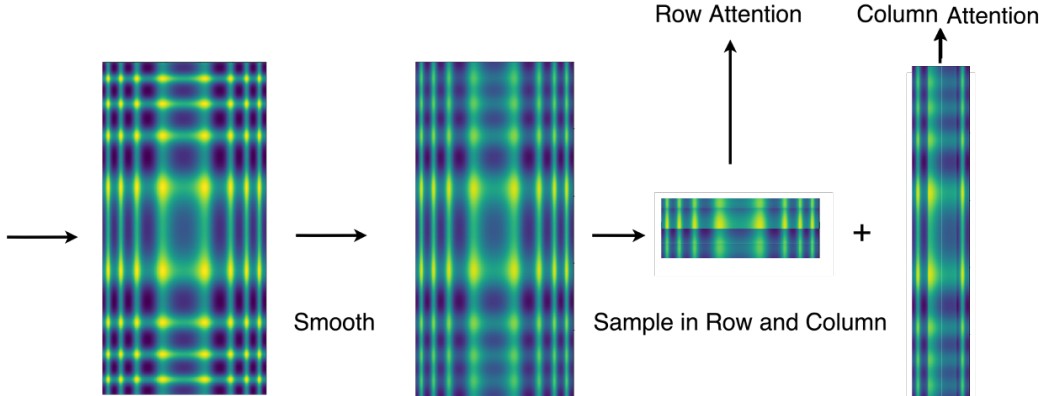

Figure 2: Illustration on effect of the Smoother and Skeleton Attention on Token Matrix.

## K    TRANSFER LEARNING

Table 15: The training configurations for Pretraining and GLUE tasks

|  | Pre-training | GLUE |
|---|---|---|
| Max Steps | 1000K | - |
| Max Epochs | - | [4,20] |
| Learning Rate | 1e-4 | [5e-5,1e-4] |
| Batch Size | 256 | [16,32] |
| Warm-up Steps | 5000 | - |
| Sequence Length | 512 | 128 |
| Learning Rate Decay | - | Linear |
| Clip | - | 1 |
| Dropout | - | 0.1 |

## L EXTRA ALGORITHMS FOR LONGTERM TIMESERIES FORECASTING

Table 16: multivariate long-term series forecasting results on six datasets with input length of 96 and prediction length $O \in \{96, 192, 336, 720\}$ (For ILI dataset, we set prediction length $O \in \{24, 36, 48, 60\}$) with input length 60. A lower MSE indicates better performance. All experiments are repeated 5 times.

| Methods | | SKTformer | | FEDformer | | Autoformer | | S4 | | Informer | | LogTrans | | Reformer | | Performer | | Nystroformer | |
|---|---|---|---|---|---|---|---|---|---|---|---|---|---|---|---|---|---|---|---|---|
| Metric | | MSE | MAE | MSE | MAE | MSE | MAE | MSE | MAE | MSE | MAE | MSE | MAE | MSE | MAE | MSE | MAE | MSE | MAE |
| ETTm2 | 96 | **0.192** | **0.283** | 0.203 | 0.287 | 0.255 | 0.339 | 0.705 | 0.690 | 0.365 | 0.453 | 0.768 | 0.642 | 0.658 | 0.619 | 2.520 | 1.220 | 0.617 | 0.600 |
| | 192 | **0.255** | **0.324** | 0.269 | 0.328 | 0.281 | 0.340 | 0.924 | 0.692 | 0.533 | 0.563 | 0.989 | 0.757 | 1.078 | 0.827 | 0.949 | 0.745 | 0.856 | 0.706 |
| | 336 | **0.324** | **0.364** | 0.325 | 0.366 | 0.339 | 0.372 | 1.364 | 0.877 | 1.363 | 0.887 | 1.334 | 0.872 | 1.549 | 0.972 | 1.701 | 1.001 | 1.394 | 0.887 |
| | 720 | 0.431 | 0.433 | **0.421** | **0.415** | 0.422 | 0.419 | 0.877 | 1.074 | 3.379 | 1.338 | 3.048 | 1.328 | 2.631 | 1.242 | 2.531 | 1.228 | 2.466 | 1.185 |
| Electricity | 96 | 0.218 | 0.332 | **0.183** | **0.297** | 0.201 | 0.317 | 0.304 | 0.405 | 0.274 | 0.368 | 0.258 | 0.357 | 0.312 | 0.402 | 0.281 | 0.375 | 0.273 | 0.364 |
| | 192 | 0.259 | 0.361 | **0.195** | **0.308** | 0.222 | 0.334 | 0.313 | 0.413 | 0.296 | 0.386 | 0.266 | 0.368 | 0.348 | 0.433 | 0.283 | 0.387 | 0.282 | 0.370 |
| | 336 | 0.267 | 0.367 | **0.212** | **0.313** | 0.231 | 0.338 | 0.290 | 0.381 | 0.300 | 0.394 | 0.280 | 0.380 | 0.350 | 0.433 | 0.301 | 0.387 | 0.302 | 0.387 |
| | 720 | 0.293 | 0.385 | **0.231** | **0.343** | 0.254 | 0.361 | 0.262 | 0.344 | 0.373 | 0.439 | 0.283 | 0.376 | 0.340 | 0.420 | 0.301 | 0.387 | 0.292 | 0.373 |
| Exchange | 96 | **0.086** | **0.204** | 0.139 | 0.276 | 0.197 | 0.323 | 1.292 | 0.849 | 0.847 | 0.752 | 0.968 | 0.812 | 1.065 | 0.829 | 0.801 | 0.729 | 0.801 | 0.729 |
| | 192 | **0.188** | **0.292** | 0.256 | 0.369 | 0.300 | 0.369 | 1.631 | 0.968 | 1.204 | 0.895 | 1.040 | 0.851 | 1.188 | 0.906 | 1.284 | 0.925 | 1.284 | 0.925 |
| | 336 | **0.356** | **0.433** | 0.426 | 0.464 | 0.509 | 0.524 | 2.225 | 1.145 | 1.672 | 1.036 | 1.659 | 1.081 | 1.357 | 0.976 | 1.408 | 0.964 | 1.408 | 0.964 |
| | 720 | **0.727** | **0.669** | 1.090 | 0.800 | 1.447 | 0.941 | 2.521 | 1.245 | 2.478 | 1.310 | 1.941 | 1.127 | 1.510 | 1.016 | 1.654 | 1.017 | 1.654 | 1.017 |
| Traffic | 96 | 0.592 | 0.352 | **0.562** | **0.349** | 0.613 | 0.388 | 0.824 | 0.514 | 0.719 | 0.391 | 0.684 | 0.384 | 0.732 | 0.423 | 0.709 | 0.391 | 0.709 | 0.400 |
| | 192 | 0.583 | 0.343 | **0.562** | **0.346** | 0.616 | 0.382 | 1.106 | 0.672 | 0.696 | 0.379 | 0.685 | 0.390 | 0.733 | 0.420 | 0.681 | 0.369 | 1.127 | 0.611 |
| | 336 | 0.598 | 0.346 | **0.570** | **0.323** | 0.622 | 0.337 | 1.084 | 0.627 | 0.777 | 0.420 | 0.733 | 0.408 | 0.742 | 0.420 | 0.682 | 0.366 | 0.867 | 0.477 |
| | 720 | 0.641 | 0.397 | **0.596** | **0.368** | 0.660 | 0.408 | 1.536 | 0.845 | 0.864 | 0.472 | 0.717 | 0.396 | 0.755 | 0.423 | 0.675 | 0.360 | 0.686 | 0.369 |
| Weather | 96 | **0.182** | **0.262** | 0.217 | 0.296 | 0.266 | 0.336 | 0.406 | 0.444 | 0.300 | 0.384 | 0.458 | 0.490 | 0.689 | 0.596 | 0.597 | 0.598 | 0.701 | 0.612 |
| | 192 | **0.228** | **0.306** | 0.276 | 0.336 | 0.307 | 0.367 | 0.525 | 0.527 | 0.598 | 0.544 | 0.658 | 0.589 | 0.752 | 0.638 | 0.606 | 0.587 | 0.655 | 0.604 |
| | 336 | **0.295** | **0.355** | 0.339 | 0.380 | 0.359 | 0.395 | 0.531 | 0.539 | 0.578 | 0.523 | 0.797 | 0.652 | 0.639 | 0.596 | 0.731 | 0.646 | 0.746 | 0.642 |
| | 720 | **0.383** | **0.418** | 0.403 | 0.428 | 0.578 | 0.578 | 0.419 | 0.428 | 1.059 | 0.741 | 0.869 | 0.675 | 1.130 | 0.792 | 0.837 | 0.682 | 0.961 | 0.751 |
| ILI | 24 | **2.185** | **0.926** | 2.203 | 0.963 | 3.483 | 1.287 | 4.631 | 1.484 | 5.764 | 1.677 | 4.480 | 1.444 | 4.400 | 1.382 | 3.937 | 1.298 | 4.378 | 1.364 |
| | 36 | **2.155** | **0.937** | 2.272 | 0.976 | 3.103 | 1.148 | 4.123 | 1.348 | 4.755 | 1.467 | 4.799 | 1.467 | 4.783 | 1.448 | 4.007 | 1.329 | 5.332 | 1.554 |
| | 48 | 2.333 | 0.954 | **2.209** | **0.981** | 2.669 | 1.085 | 4.066 | 1.36 | 4.763 | 1.469 | 4.800 | 1.468 | 4.832 | 1.465 | 4.575 | 1.451 | 5.575 | 1.614 |
| | 60 | **2.018** | **0.958** | 2.545 | 1.061 | 2.770 | 1.125 | 4.278 | 1.41 | 5.264 | 1.564 | 5.278 | 1.560 | 4.882 | 1.483 | 4.020 | 1.366 | 4.742 | 1.469 |

