# OpenReview forum: "SKTformer: A Skeleton Transformer for Long Sequence Data"
_ICLR.cc/2023/Conference — Submitted to ICLR 2023_

### Official Review · Reviewer_6UUx · 2022-10-23

**Confidence:** 3
**Correctness:** 3
**Technical Novelty And Significance:** 3
**Empirical Novelty And Significance:** 3
**Recommendation:** 6

**Clarity, Quality, Novelty And Reproducibility:**

The paper is well-written overall. It might be better to re-organize the theoretical analysis in a more coherent way, apart from the model architecture, for better readability. Regarding reproducibility, the code is publically available, and implementation details are also provided in the supplementary materials.

**Strength And Weaknesses:**

[+] This paper proposes a novel idea for modeling long sequences and its performance is well-validated through extensive experiments.

[+]  The idea of using a subset of rows or columns for self-attention and combining both is interesting. It reduces computational complexity and also improves overall performance. It seems meaningful that the proposed model is not only computationally efficient but also robust against noise.


[-] The choice of models for comparison is not consistent across Tables. For example, Nystroformer and Performer show good train speeds in Table 4 but are not included in Table 2. There is a tradeoff between performance and computational complexity, so a clearer explanation of why the chosen methods are used in each experiment will make the comparison fairer.

[-] Because the advantage of this model is that it has high computational efficiency as well as good performance, I think it is good to add models with higher performance than the models proposed in the training speed experiment as a baseline model.

[-] The description of the theoretical analysis could be improved. For example, Lemma 1 is included in the last part of section 3.1: skeleton attention, but it’s not well explained in the context of 3.1. In addition, since the concept of mu-incoherence might not be familiar to some readers, it’d be better to introduce it near Lemma 1 in the same section, before using it in a later part. It's defined in the supplementary but not mentioned in the main manuscript.

[-] Table 6 is not referred to elsewhere in the manuscript.

[Q]. The choice for the value of hyperparameters such as s1 and s2 in combination with other hyperparameters seems to affect the performance much. Is there a guideline or suggestion about an efficient search strategy?

[Minor]
- In abstract: villain -> vanilla
- Section 4.4, 2nd line: SKTformer achieve -> SKTformer achieves
- Section 5, 1st line: n robust and efficient transformer architecture -> a robust and efficient transformer architecture




**Summary Of The Paper:**

This paper proposes an efficient transformer architecture, a skeleton transformer (SKTformer), for modeling long sequence data. It contains two main components: a smoothing block to mix information over long sequences through Fourier convolution, and a matrix sketch method that simultaneously selects columns and rows from the input matrix for efficient self-attention. Theoretical and experimental analyses of the proposed method are provided and its performance is validated compared to other Transformer variants on several tasks.

**Summary Of The Review:**

The novelty of the proposed method for efficient self-attention attention and other mechanisms introduced could inspire the readers. Also, the authors have conducted many experiments for validation. My concern is mostly on the selection of comparison methods shown in some of the experiments.  If these concerns can be resolved, I think it's worth being introduced at this conference.

---

### Official Review · Reviewer_9G8D · 2022-10-23

**Confidence:** 4
**Clarity, Quality, Novelty And Reproducibility:** See clarity issues above. With unclea…
**Correctness:** 2
**Technical Novelty And Significance:** 2
**Empirical Novelty And Significance:** 2
**Recommendation:** 6

**Strength And Weaknesses:**

Clarity is a major concern. The theoretical arguments are decorative but of little relevance and significance. In fact, the lemmas harm clarity. They do not help answer the key question: "why should we smooth the key matrices?" We all know that smoothed matrices are easier to approximate via sampling. However, after all, approximating the matrix is not the purpose of learning representations.

Even if we assume smoothing is really helpful, there is still major unclearness in method development. Section 3.2.1 aims to design an efficient smoothing method. However, it is very hard to read. In specific, I don't see why Eq (5) can approximate Eq (4). The $S$ matrix in Eq. (5) essentially applies an averaging pooling mask along the feature dimension. Its effect is unexplained. Note that lemma 2 and 3 do not provide an answer.

Also, the smoothing is applied to the input, whereas eventually we need smoothed keys and values. There are linear transforms that map input to keys and values. does smoothed input necessarily lead to smoothed keys and values?

Minor: Lemmas are for proving major theorems. But here the authors mainly want to argue for correctness. I'd suggest change "lemma" to "proposition".

**Summary Of The Paper:**

Through multiple iteration of rebuttal, the authors have provided empirical evidence to address my concern. I am willing to raise my score to 6, conditioned on polishing the method section. The following are my original review:

This paper proposes to sample rows and columns of key and query matrices to approximate attention matrix in transformer layers. The authors argue that if these matrices are smooth enough, then the downsampled matrix should retain most information. Therefore, the authors propose to smooth input by convolving it with some learnable kernel.

**Summary Of The Review:**

This paper has major clarity issues and cannot justify its correctness.

---

### Official Review · Reviewer_ELnS · 2022-10-25

**Confidence:** 3
**Correctness:** 3
**Technical Novelty And Significance:** 3
**Empirical Novelty And Significance:** 3
**Recommendation:** 6

**Clarity, Quality, Novelty And Reproducibility:**

This writing is clear in general.

The quality is good.

The skeleton attention and smooth convolution are novel.

The reproducibility is good since implementation details are well-documented.

**Strength And Weaknesses:**

Strengths:
1. The proposed Skeleton attention and Fast Fourier Transformation based smooth convolution is novel.
2. Theoretical analysis is provided to demonstrate SKTformer could reduce complexity and retain sufficient information.
3. The experimental results show that SKTformer could outperform SOTA methods to a certain degree.

Weaknesses:
1. It is necessary to perform t-tests for SKTformer since it does not consistently outperform SOTA methods.

**Summary Of The Paper:**

This paper introduces a novel SKTformer for long sequence modeling, which reduces the complexity of the vanilla transformer based on the Skeleton approximation. There are two major novelties: Skeleton attention and Fast Fourier Transformation-based smooth convolution. Theoretically, analysis is provided to show the proposed model could reduce the complexity. The experimental results show that SKTformer could outperform existing methods to a certain degree.

**Summary Of The Review:**

In general, this paper introduces a novel SKTformer to reduce the complexity of the Transformer. Theoretical analysis is provided and the empirical results demonstrate the effectiveness of SKTformer. However, SKTformer does not consistently outperform SOTA baselines. Thus, it is necessary to conduct t-tests.

---

### Official Review · Reviewer_FKSC · 2022-10-26

**Confidence:** 3
**Correctness:** 3
**Technical Novelty And Significance:** 3
**Empirical Novelty And Significance:** 3
**Recommendation:** 6

**Clarity, Quality, Novelty And Reproducibility:**

** Clarity **

Generally, the paper is well organized and is well-written.

It does have a surprising amount of typos, but those can be easily resolved.

Additionally, sometimes the phrasing is a little suboptimal. For example, in the abstract, the authors say that their method "addresses" the tradeoff between computational cost, information retention, and noise. It's not clear what is meant by that. The paper does not propose a method to avoid this tradeoff completely; it might improve upon previous results, but, in my understanding, it does not address the tradeoff. Perhaps something along the lines of "improves upon the previous results/attempts to negotiate this tradeoff" would work better in this context.

Additionally, table numeration in appendix is very confusing. Specifically, table 10 is listed in text noticeably later than table 11. I suggest that tables are re-numbered.

Overall, as long as these issues (especially typos) are resolved, the clarity matches the standards of the ICLR conference. I trust the authors to fix simple typos, so they did not affect my overall assessment.

** Quality **

The experiments are well - designed and well-executed. I find it preferable to highlight the fixed hyperparameter SKTformer (r, s1, s2 = 8) performance more, since this is more consistent with how results were previously reported in the literature (e.g. in the Long Range Arena,. and in Luna: Linear Unified Nested Attention, the authors do not fine-tune major hyperparameters to each subtask).

Additionally, run-to-run variability is not included. Since even for the fixed hyperparameter model, some hyperparameter search was still performed (number of epochs, batch size, dropout, etc.), it's highly important to report results averaged over different runs & their variability. Methodologically, this is my main concern.

To be precise, it is reported, but hidden. It's only in the "Appendix G" that the authors mention the run-to-run variability and the fact that the results they report are averaged over 5 runs. This information (and the variability of the overall result, not only of each separate task) should be included in Table 1.

** Novelty **

The novelty/originality of this paper is not its main selling point, as the method is largely a combination of pre-existing techniques, but it is sufficient & up to the standards of the ICLR conference.

** Reproducibility **

As far as I can judge, the detailed model description & supplementary materials make this paper comply with highest reproducibility standards.

** Typos and other suggestions **

Generally, the amount of typos in this paper is a little concerning (I provide a sample below). I would suggest a thorough proofread of the paper before the final version is submitted.

Extensive studies over both Long Range Arena (LRA) datasets, six time-series forecasting show that SKTformer significantly outperforms ->
..., and six time-series

without having to suffice the linear complexity w.r.t. sequence length ->  without having to sacrifice the linear complexity w.r.t. sequence length

We propose SKTformer, n robust -> We propose SKTformer, a robust

build the column self-attention as follow -> build the column self-attention as follows

**Strength And Weaknesses:**

Strengths:
- The paper addresses a highly important problem
- The proposed method shows good results on a sufficiently wide range of problems
- Experiments are thorough, and ablation results are reported
- The paper is generally clearly organized

Weaknesses:
- The most impressive results are achieved when hyper parameters of the proposed method are freely varied for each subtask. It may be creating an overly optimistic impression of the model performance. I'd prefer a higher emphasis on the performance of the fixed hyperparameter version (i.e. SKTformer (r, s1, s2 = 8)). For the benchmark used (Long Range Arena), it goes contrary to the established approach.
- Performance variability from run to run is not reported fully (its reported in the appendix for LRA substasks, but not for the overall performance, and is not referenced in the main text).

**Summary Of The Paper:**

The paper proposes a new approach to apply transformers to Long Sequence Data, called SKTFormer. The approach combines a CUR matrix approximation technique, a Fourier-convolution based smoother, and a convolution stem to avoid over-smoothing. The authors apply their method to a number of problems, showing promising performance; they also report ablation study results, analyzing the importance of different components of their method.

**Summary Of The Review:**

Generally, the paper addresses an important problem, and runs a number of reasonably planned experiments, supplementing them with theoretical results.

Its main issue is the way experimental results are reported. Contrary to how most previous papers approached the topic (see Long Range Arena original paper, section 3.2 "Philosophy Behind the Benchmark"), authors fine-tuned the major hyperparameters for each subtask in the Long Range Arena. Given that these hyperparameters do have a major impact on performance (see table 7), such reporting may create a confusing picture of how good the method actually is, and make it harder to compare it with both previous and future results. The authors report a fixed hyperparameter version, but only as a secondary metric, and I suggest that it is reported more clearly either along or instead of the fine-tuned one, e.g. in the abstract and other parts throughout the paper.

Additionally, even though Appendix G gives the performance variability for each subtask in LRA, it is not mentioned in the main text and does not give the variability of the overall result, which is the most important part (I strongly suggest that it's included for reader's convenience, especially since it can be calculated assuming independence at no additional computational cost).

Overall, however, the pros outweigh the cons, and I believe that the paper is above the acceptance threshold, although, unless the issues above are fixed, I believe that it's borderline. I will stay open to adjust my assessment based on the rebuttal and other reviewer's comments.

** UPD **

I have read other reviews and the authors' responses. They alleviate many of the concerns voiced by me and other reviews. In my opinion, I can not quite improve the score to an "8", but I would have given the paper a "7" if I could.

---

### Decision · Program_Chairs · 2023-01-20

**Decision:**

Reject

**Justification For Why Not Higher Score:**

N/A

**Justification For Why Not Lower Score:**

N/A

**Metareview: Summary, Strengths And Weaknesses:**

The paper proposes Skeleton Transformer (SKTformer), a model that tries to balance the trade off between memory and noise reduction.

Strength:
- The model outperforms on some tasks many Transformer baselines.
- The paper is clearly written and well organized.

Weaknesses:
- The paper starts by stating an outdated statement: "Transformers have become a preferred tool for modeling sequential data". This is an inaccurate statement by large. The Long Range Arena which comprises one of the primary benchmarks of this paper has been dominated by state-space models and variants by a large margin. Unfortunately, the authors did not even include a comparison with these state-of-the-art models for long-term dependencies such as the non-exhaustive list below:

[1] Gu, A., Goel, K., & Ré, C. (2021). Efficiently modeling long sequences with structured state spaces. arXiv preprint arXiv:2111.00396

[2] Gupta, A. (2022). Diagonal State Spaces are as Effective as Structured State Spaces. arXiv preprint arXiv:2203.14343.

[3] Gu, A., Johnson, I., Goel, K., Saab, K., Dao, T., Rudra, A., & Ré, C. (2021). Combining recurrent, convolutional, and continuous-time models with linear state space layers. Advances in neural information processing systems, 34, 572-585.

[4] Gu, A., Dao, T., Ermon, S., Rudra, A., & Ré, C. (2020). Hippo: Recurrent memory with optimal polynomial projections. Advances in Neural Information Processing Systems, 33, 1474-1487.

[5] Smith, J. T., Warrington, A., & Linderman, S. W. (2022). Simplified state space layers for sequence modeling. arXiv preprint arXiv:2208.04933.

[6] Hasani, R., Lechner, M., Wang, T.H., Chahine, M., Amini, A. and Rus, D., 2022. Liquid structural state-space models. arXiv preprint arXiv:2209.12951.

[7] Gu, A., Gupta, A., Goel, K. and Ré, C., 2022. On the parameterization and initialization of diagonal state space models. arXiv preprint arXiv:2206.11893.

[8] Gu, A., Johnson, I., Timalsina, A., Rudra, A. and Ré, C., 2022. How to train your hippo: State space models with generalized orthogonal basis projections. arXiv preprint arXiv:2206.12037

- The entire experiments on time-series forecasting (Section 4.2) are done on short-term sequences (below 720 steps) where even an ARIMA model, XGBoost, or a well-tuned LSTM could perform competitively. The use of a Transformer architecture on these datasets is not appropriate at all.

- There are established and challenging benchmarks with 8k-16k samples in the time-series domain. An example is the Speech Command recognition dataset used in all modern sequence modeling works. Transformers suffer from learning such long time series due to computational complexity. They suffer even more when they are trained on sequences sampled at 16 kHz and then tested on sequences with 8 kHz. This is the kind of generalization one would ideally expect from a sequence modeling tool. In this space, RNNs and State-space models work remarkably well.

- The state-of-the-art Transformer model (Mega) [Ma et al. 2022] on sequence modeling tasks has not been included in the analysis of the paper. This is especially important as the Mega model outperforms all transformers including SKTformer by a significant margin. Please see: https://arxiv.org/pdf/2209.10655.pdf

Overall, after discussions with the reviewer, we found the impact of the work minimal and of incremental value. I vote for rejection and encourage the authors to work on the suggestions made to make substantial improvements to their future work.



**Summary Of Ac-Reviewer Meeting:**

We met with the reviewers and had the following conclusions:

Concerns:
- The work is incremental.
- The technique is useful, but not necessarily increase our understanding.
- Math should be better explained
- The state-of-the-art sequence models have not been included in the experimental results of the work.